# Ice Sheet Model Intercomparison Project (ISMIP6) contribution to CMIP6

Sophie M.J. Nowicki[1], Tony Payne[2], Eric Larour[3], Helene Seroussi[3], Heiko Goelzer[4,5], William Lipscomb[6], Jonathan Gregory[7,8], Ayako Abe-Ouchi[9,10], and Andrew Shepherd[11]

[1]NASA Goddard Space Flight Center, Greenbelt, MD 20771, USA
[2]School of Geographical Sciences, University of Bristol, Bristol, BS8 1SS, UK
[3]Jet Propulsion Laboratory, California Institute of Technology, Pasadena, CA 91109, USA
[4]Institute for Marine and Atmospheric Research, Utrecht University, Utrecht, 3584 CC, NL
[5]Laboratoire de Glaciologie, Université Libre de Bruxelles, CP160/03, Av. F. Roosevelt 50, 1050 Brussels, BE
[6]Los Alamos National Laboratory, Los Alamos, NM 87544, USA
[7]Department of Meteorology, University of Reading, Reading, RG6 6BB, UK
[8]Met Office Hadley Center, Exeter, EX1 3BP, UK
[9]Atmosphere and Ocean Research Institute, The University of Tokyo, Kashiwa-shi, Chiba 277-8564, JP
[10]Japan Agency for Marine-Earth Science and Technology, Yokohama, JP
[11]School of Earth and Environment, University of Leeds, Leeds, LS2 9JT, UK

*Correspondence to*: Sophie M.J. Nowicki (sophie.nowicki@nasa.gov)

**Abstract.** Reducing the uncertainty in the past, present and future contribution of ice sheets to sea-level change requires a coordinated effort between the climate and glaciology communities. The Ice Sheet Model Intercomparison Project for CMIP6 (ISMIP6) is the primary activity within the Coupled Model Intercomparison Project – phase 6 (CMIP6) focusing on the Greenland and Antarctic Ice Sheets. In this paper, we describe the framework for ISMIP6 and its relationship to other activities within CMIP6. The ISMIP6 experimental design relies on CMIP6 climate models and includes, for the first time within CMIP, coupled ice sheet – climate models as well as standalone ice sheet models. To facilitate analysis of the multi-model ensemble and to generate a set of standard climate inputs for standalone ice sheet models, ISMIP6 defines a protocol for all variables related to ice sheets. ISMIP6 will provide a basis for investigating the feedbacks, impacts, and sea-level changes associated with dynamic ice sheets and for quantifying the uncertainty in ice-sheet-sourced global sea-level change.

## 1    Introduction

Ice sheets constitute the largest and most uncertain potential source of future sea-level rise (Church et al., 2013, Kopp et al., 2014). The Greenland and Antarctic Ice Sheets currently hold ice equivalent of over 7 and 57 meters of sea-level rise, respectively. Observations indicate that the Greenland and Antarctic Ice Sheets have contributed approximately 7.5 mm and 4 mm of sea-level rise over the 1992-2011 period (Shepherd et al., 2012) and that their contribution to sea-level rise is accelerating (Rignot et al., 2011a). Sea-level change has been identified as a long-lasting consequence of anthropogenic

climate change, as sea levels will continue to rise even if temperatures are stabilized (Meehl et al., 2012). Therefore, assessing whether the observed rate of mass loss from the ice sheets will continue at the same pace, or accelerate, is crucial for risk assessment and adaptation efforts.

In addition to their impact on sea-level change, ice sheets influence the Earth's climate through changes in freshwater fluxes, orography, surface albedo and vegetation cover, across multiple spatial and temporal scales (Vizcaíno, 2014). Ice-sheet evolution and iceberg discharge affect ocean freshwater fluxes (e.g., Broecker, 1994), which in turn can affect oceanic circulation (e.g., Weaver et al., 2003), and marine biogeochemistry (Raiswell et al., 2006). Changes in ice sheet orography modify near-surface temperatures by altering atmospheric circulation (Ridley et al., 2005) on both regional and global scales
(e.g., Manabe and Broccoli, 1985). Surface albedo and elevation change due to the waxing and waning of ice sheets has played an important role in past interglacial-glacial transitions (e.g., Calov et al., 2009; Abe-Ouchi et al., 2013). Seasonal fluctuations in ice-sheet albedo can also exert considerable influence on local surface energy fluxes (e.g., Box et al., 2012), through both melt and snowfall. Over longer timescales, changes in ice-sheet elevation can cause a positive feedback on surface mass balance, wherein a thinning ice sheet experiences warmer temperatures at lower elevations, which causes
further melting and thinning. Ice-sheet elevation changes can also alter the local climate, for instance changing the trajectory of Southern Ocean storms that penetrate onto the Antarctic Plateau (Morse et al., 1998).

Ice sheets gain mass primarily by accumulation of snowfall, and lose mass through a combination of surface meltwater runoff, surface sublimation, iceberg discharge to the ocean, and basal melting (under both grounded ice and floating ice
shelves). The Antarctic Ice Sheet experiences minimal surface melt and thus loses mass primarily through basal melting and iceberg calving. Most basal mass loss in Antarctica occurs under ice shelves (e.g. Joughin et al., 2003; Pritchard et al., 2012), but sub-ice-sheet meltwater is also produced over large areas (Fricker et al., 2007). Together, basal melting and iceberg calving currently outweigh snowfall accumulation to the Antarctic Ice Sheet (Rignot et al., 2013; Depoorter et al., 2013). The Greenland Ice Sheet is also currently losing mass overall; this occurs primarily through iceberg calving and
surface runoff. Surface mass balance changes have recently surpassed iceberg calving changes as the dominant contributor to Greenland mass loss (van den Broeke et al., 2009), with increased surface runoff now contributing 60% of the mass loss (Enderlin et al., 2014). Due to the long response time of ice sheets, mass changes observed at present are a complex combination of the response to present climate changes, as well as past climate changes as far back as several tens of thousands of years. These integrating effects of ice sheets and the vastly different time scales on which ice sheet models and
climate models operate have historically inhibited efforts to interface these two components of the Earth system.

Previously, ice sheets were not explicitly included in the CMIP process, and separate modeling studies were used to make projections of their future contributions to sea level. This has often led to mismatches between the climate data used to force these models and the contemporary version of the CMIP projections. This mismatch was perhaps acceptable when ice sheets

were regarded as passive elements of the climate system on sub-millennial time scales (e.g., Church and Gregory, 2001). Observations of rapid mass loss associated with dynamic change in the ice sheets, however, have highlighted the need to couple ice sheets to the rest of the climate system. At one stage, this mismatch was such that little confidence could be placed in the projections of ice-sheet models, which were felt to omit the key processes responsible for observed changes (e.g., Meehl et al., 2007). With subsequent developments in ice-sheet modeling, many of the processes thought to affect ice-sheet dynamics on sub-centennial time scales (such as grounding-line migration, changes in basal lubrication and to some extent iceberg calving) can be simulated with some confidence (e.g., Church et al., 2013). Previous ice sheet model inter-comparison exercises have played a crucial role in this development. An excellent example is the ongoing series of inter-comparisons aimed at understanding issues associated with the numerical modeling of grounding-line motion (e.g., Pattyn et al., 2012, 2013). Two previous international efforts, the SeaRISE and ice2sea initiatives, supplied projections on which the assessments of Church et al. (2013) were based. A major criticism of both efforts, however, was that they were based on forcing from the Special Report on Emissions Scenarios (SRES, Nakićenović et al., 2000) rather than the current Representative Concentration Pathway (RCP, van Vuuren et al., 2011) framework. The Ice Sheet Model Intercomparison Project for CMIP6 (ISMIP6) is explicitly designed to ensure that ice sheet (hence sea-level) projections are fully compatible with the CMIP6 process.

ISMIP6 brings together for the first time a consortium of international ice sheet models and coupled ice sheet – climate models. This effort will thoroughly explore the sea-level contribution from the Greenland and Antarctic Ice Sheets in a changing climate and assess the impact of large ice sheets on the climate system. In this paper, we provide an overview of the ISMIP6 effort and present the ISMIP6 framework. We begin by explaining the objectives and approach for ISMIP6 (Sect. 2), and describe the experimental design (Sect. 3). We next present an evaluation and analysis plan (Sect. 4) and finally discuss the expected outcome and impact of ISMIP6 (Sect. 5).

## 2   Objectives and Approach

ISMIP6 was initiated with the help of the Climate and Cryosphere (CliC) effort of the World Climate Research Project (WCRP) and is now a targeted activity of CliC. The main goal is to better integrate ice sheet models in climate research in general, and in the CMIP initiative in particular. ISMIP6 offers the exciting opportunity of widening the current CMIP definition of the Earth System to include ice sheets. Together with the CliC targeted activity on glacier modeling (GlacierMIP) and existing models for thermal expansion within the CMIP framework, output from ISMIP6 will add sea level to the family of variables for which CMIP can provide routine IPCC-style projections. ISMIP6 is primarily focused on the CMIP6 scientific question "How does the Earth System respond to forcing?", but will also contribute to answering the question "How can we assess future climate change given climate variability, climate predictability and uncertainty in climate scenarios?" for scenarios involving the mass budget of the ice sheets and its impact on global sea level.

ISMIP6 targets two Grand Science Challenges (GCs) of the WRCP: "Melting Ice and Global Consequences" and "Regional Sea-level Change and Coastal Impacts". Specifically, the primary goal of the ISMIP6 effort is to improve our understanding of the evolution of the Greenland and Antarctic Ice Sheets under a changing climate. A related goal is to quantify past and future sea-level contributions from ice sheets, including the associated uncertainties. These uncertainties arise from uncertainties in both the climate input and the response of the ice sheets. A secondary goal is to investigate the role of feedbacks between ice sheets and climate in order to gain insight into how changes in the ice sheets will affect the Earth climate system.

These goals require an experimental framework that can address the following objectives:

- Develop better models of climate and ice sheets, as both coupled systems and individual components
- Improve understanding of how ice sheets respond to climate on various timescales, both in the past and in the future
- Improve understanding of how ice sheets affect local and global climate, and explore ice sheet-climate feedbacks
- Improve simulation of sea-level change, especially projections for the 21$^{st}$ century and over the next 300 years

As depicted in Fig. 1, our goals and objectives rely on three distinct modeling efforts: i) traditional CMIP atmosphere – ocean general circulation models (AOGCM/AGCMs) without dynamic ice sheets, ii) standalone dynamic ice sheet models (ISMs) that are driven by provided forcing fields ("offline"), and iii) atmosphere-ocean climate models coupled to dynamic ice sheets (AOGCM-ISMs), which, as described in the following sections, can be combined to form an integrated framework.

### 3    ISMIP6 Experimental Design

Following the CMIP6 protocol, the ISMIP6 experiments both use and augment the CMIP6-DECK (Diagnostic Evaluation and Characterization of Klima) and Historical simulations (Meehl et al., 2014; Eyring et al., 2016). In addition, ISMIP6 collaborates with the CMIP6-Endorsed Paleoclimate Model Intercomparison effort (PMIP4, Kageyama et al., 2016) and builds on the CMIP6-Endorsed ScenarioMIP (O'Neill et al., 2016) that focuses on future climate experiments for CMIP6. For a selected number of AGCM/AOGCM experiments that are already part of CMIP6 (Table 1 and described in Sect. 3.1), three additional model configurations are proposed: "*XXX-withism*", "*ism-XXX-self*" and "*ism-XXX-std*", where XXX stands for different forcing scenarios as described later and shown in Table 2. The first case, "*XXX-withism*", indicates that the ice sheet model is run interactively with the climate model (the AOGCM-ISM configuration described in Sect. 3.2). The other two cases describe an offline, or "standalone", ice sheet model that is driven by outputs from either an uncoupled AOGCM "*ism-XXX-self*" (the ISM configuration described in Sect. 3.2) or from a standard ISMIP6 dataset "*ism-XXX-std*" that will be provided for the glaciology community (the ISM configuration described in Sect. 3.3). The goal of the *ism-XXX-self*

simulations is to obtain an ice sheet evolution and sea-level contribution that can be compared to the AOGCM-only and the AOGCM-ISM experiments in order to gain insight into the feedbacks between ice sheets and climate. Differences between the *ism-XXX-self* runs and AOGCM-ISM runs will be attributable to ice-sheet feedbacks on other climate components. The *ism-XXX-std* experiments will complement the AOGCM and AOGCM-ISM experiments by using ice sheet configurations

and forcing data sets that are as realistic as possible, aiming to minimize the effects of AOGCM biases. The *ism-XXX-std* simulations target mainly the glaciology community and aim to simulate realistic ice-sheet evolution for sea-level estimates. A related set of standalone experiments, called initMIP, will explore uncertainties associated with the initialization of ice sheet models for Greenland and Antarctica.

### 3.1 Analysis of experiments with climate models proposed elsewhere in CMIP6 (and not coupled to ISMs)

A first component of the ISMIP6 effort is to assess and evaluate CMIP atmosphere general circulation models (AGCMs) and coupled atmosphere – ocean general circulation models (AOGCMs) over and surrounding the polar ice sheets. This part of ISMIP6 can be viewed as diagnostic in the sense that all climate models that participate in CMIP6 will be included in this assessment without requiring extra work from the climate modeling centers. These experiments do not include dynamic ice sheets, and as explained in the CMIP6 protocol (Eyring et al., 2016), climate modeling centers that contribute to CMIP6 are

required to submit simulations for the DECK and CMIP6 Historical runs. Our goals are to establish the suitability of the CMIP models for producing climate input for ice sheet models and to assess the uncertainty in projections of sea-level change arising from such climate input. As described in Sect. 4, an additional goal is to assess past and projected changes in surface forcing (here for a fixed ice-sheet extent and topography), along with the resulting sea-level contribution from both ice sheets due to changes in surface freshwater flux alone. The largest uncertainty in century-scale sea-level projections,

however, remains the dynamic ice sheet response to changes in atmospheric and oceanic conditions, which will be addressed by the other components of ISMIP6 (Sect. 3.2 and 3.3).

The experiments with climate models not coupled to ISMs, listed in Table 1, are central to ISMIP6 and thus briefly introduced. These AGCM/AOGCM experiments are already part of CMIP6, such that more detailed information on the

experimental protocol is available elsewhere in this special issue. ISMIP6 uses three of the four DECK experiments described in Eyring et al. (2016).The Atmospheric Model Intercomparison Project (*amip,* Gates et al., 1999) simulation allows the evaluation of the atmospheric component of climate models given prescribed sea-surface temperatures and sea ice conditions. These oceanic forcings are based on observations and range from January 1979 to December 2014 for CMIP6 (see Appendix A1.1 of Eyring et al., 2016). The pre-industrial control, *piControl*, is a coupled atmospheric and oceanic

simulation with constant conditions, chosen to represent pre-industrial values (with 1850 as the reference year, see Appendix A1.2 of Eyring et al., 2016). *piControl* serves as the starting point for many simulations and is meant to capture the pre-industrial quasi-equilibrium state of the climate system. It allows an evaluation of model drift and provides insight into the

unforced internal variability. The DECK also contains two idealized "climate change" experiments, in which the $CO_2$ concentration is varied to gain insight into the Earth system response to basic greenhouse gas forcing. ISMIP6 will focus on a *1pctCO2to4x* simulation, a slightly modified version of the DECK *1pctCO2* simulation. The *1pctCO2* simulation is 150 years long, starting from the *piControl*, with a 1% per year increase in atmospheric $CO_2$ concentration. The *1pctCO2to4x* simulation is identical to *1pctCO2* for the first 140 years, at which point the $CO_2$ concentration reaches four times the initial value. At this point, *1pctCO2to4x* branches from *1pctCO2* and continues with constant quadrupled $CO_2$. (Note that the *1pctCO2to4x* scenario was called *1pctCO2* in CMIP5 (Taylor et al., 2012) and *1pctto4x* in CMIP3.) In order to produce boundary conditions for their *ism-1pctCO2to4x-self* simulation, groups participating in ISMIP6 with a coupled AOGCM-ISM should carry out a *1pctCO2-4xext* simulation, which starts from year 140 of their *1pctCO2* simulation and runs for a minimum of 210 years (and ideally 360 years, see Sect 3.2) with $CO_2$ concentration held fixed. The *1pctCO2to4x* fields will not be stored in the CMIP6 archive, but can be generated by merging the outputs from the first 140 years of the *1pctCO2* run with that from *1pctCO2-4xext.*

The CMIP6 Historical simulation, *historical,* tests the capability of AOGCMs to simulate the historical period, defined as 1850 to 2014. The forcing is derived from observations of solar variability and changes in atmospheric composition, including both anthropogenic and volcanic sources (see Appendix A2 of Eyring et al., 2016). The more distant past is the focus of PMIP4, which designs paleoclimate experiments (Kageyama et al., 2016; Otto-Bliesner et al., 2016). ISMIP6 collaborates with PMIP4 for experiment *lig127k*, a simulated time slice of the Last Interglacial (LIG): the warm period from 129,000 to 116,000 years ago when global mean sea level was 5–10 m higher than present (Masson-Delmott et al., 2013). The future in CMIP6 falls under the guidance of ScenarioMIP (O'Neill et al., 2016), ISMIP6 will focus on the high-emission scenario *ssp585* that produces a radiative forcing of 8.5 W m$^{-2}$ in 2100 and its extension to 2300, to evaluate climate and ice sheet changes in response to a large forcing. If time permits, lower-emission mitigation scenarios will also be included in the ISMIP6 standalone ice sheet framework.

Evaluation of the climate over and surrounding the ice sheets is necessary both to establish the suitability of current climate models to provide forcing for ice sheet models, and to gain insight into sea-level uncertainty arising from uncertainty in atmospheric and oceanic climate forcings. Of particular interest is the surface climate over the ice sheets, with a focus on temperature and surface mass balance (SMB). SMB is defined as total precipitation minus evaporation, sublimation and surface runoff, where runoff is meltwater less any refreezing within the snowpack. Because the ocean condition is prescribed for the *amip* simulation but not for the *historical* simulation, we expect that the temperature and SMB provided by the two simulations over the same time period will differ. We will explore our second interest, the capability of climate models to reproduce the oceanic state in the vicinity of the ice sheets, using the *historical* simulation.

The general approach for evaluating the atmospheric component of climate models over the ice sheets (e.g., Yoshimori and Abe-Ouchi, 2012; Fettweis et al., 2013; Vizcaíno et al., 2013; Cullather et al., 2014; Lenaerts et al., 2016) is to compare the large-scale atmospheric state over the polar regions, the local climate, and processes at the ice-sheet surface. The latter focuses on whether the climate model can simulate snow processes, including albedo evolution and refreezing, at a horizontal resolution that captures the SMB gradients at ice sheet margins. Both the atmospheric components and factors that can affect atmospheric processes are often evaluated. One example is determining whether sea ice conditions are adequately captured in *historical* simulations (e.g., Lenaerts et al., 2016), as sea ice can influence moisture availability and therefore precipitation. However, adequate modeling of precipitation also requires well-resolved ice sheet topography (orographic forcing), which remains challenging for coarse-resolution climate models (Vizcaíno, 2014).

The large-scale atmospheric state over the polar regions is often assessed by comparing the modeled atmospheric flow at 500 hPa to atmospheric reanalysis values. For the local climate, near-surface winds and near-surface temperatures can be compared to regional climate models (RCM) such as RACMO2 (van Meijgaard et al., 2008; Lenaerts et al., 2012; van Angelen et al., 2014), MAR (Fettweis, 2007; Fettweis et al., 2011), or HIRHAM (Langen et al., 2015; Lucas-Picher et al., 2012), reanalysis (e.g., Agosta et al., 2015), and observations where available. RCMs are also used to evaluate the spatial pattern of surface mass balance and its components (precipitation, sublimation, and surface melt) computed by global circulation models. The surface energy budget, particularly the seasonal cycle of net shortwave and longwave radiation and the sensible and latent heat fluxes, can be evaluated against measurements taken by automatic weather stations on the ice sheet surface. Such stations include, for example, the 15 Greenland stations known as the GC–Net (Steffen and Box, 2001), the Greenland PROMICE network with a focus on the ablation zone (Ahlstrom et al., 2008), and in Antarctica the Neumayer Base (Lenaerts et al., 2010). These stations also record winds and temperatures. The surface temperature over the ice sheets may also be evaluated from satellite observations, using, for example, data derived from the Moderate Resolution Imaging Spectroradiometer (MODIS, Hall et al., 2012). These remotely sensed temperature products show the onset and/or spatial extent of surface melt (e.g., Mote et al., 1993; Hall et al., 2013), which can then be used to assess whether the climate models capture the relevant processes at the ice sheet surface (e.g., Fettweis et al., 2011; Cullather et al., 2016). However, a full understanding of why surface melt varies from model to model may require investigations that include cloud properties (van Tricht et al., 2016).

The current generation of climate models participating in CMIP6 is unlikely to simulate ocean circulation in ice shelf cavities or within fjords. Thus, evaluation of the ocean state around the ice sheets involves first establishing that the climate models can reproduce certain properties of the key water masses. Ocean circulation around the Greenland Ice Sheet involves a complex interaction between polar waters of Arctic origin and Atlantic waters from the subtropical North Atlantic (Straneo et al., 2012). The mechanisms that transport warm water through fjords and toward the ice fronts remain an active area of research (Wilson and Straneo, 2015; Straneo and Cenedese, 2015). In the Southern Ocean, important water masses include

Antarctic Bottom Water and Antarctic Intermediate Waters. In the coastal regions, Circumpolar Deep Water, Antarctic Surface Water, and High Salinity Shelf Water are the primary oceanic influences on ice sheets (Bracegirdle et al., 2016). Given the difficulty many CMIP5 models had in capturing high-latitude ocean properties, CMIP6 models should be evaluated using existing datasets (Bracegirdle et al., 2016). These datasets include Argo, expendable bathythermograph (XBT) and conductivity/temperature/depth (CTD) vertical temperature and salinity profiles (e.g., Dong et al., 2008), sea ice extent products sourced from passive microwave instruments (e.g., Bjorgo et al., 1997; Cavalieri and Parkinson, 2012; Parkinson and Cavalieri, 2012), sea surface temperature (SST) from WindSat and AMSR-E over the open ocean, satellite altimetry (Jason-1 and Jason-2) over the open ocean, and World Ocean Atlas 2009 climatological temperatures. For ocean models that include ice-shelf cavities and ice/ocean interactions, sub-ice-shelf basal melting can be compared with glaciological estimates of ice-shelf melting around Antarctica (Rignot et al., 2013; Depoorter et al., 2013) derived from remote-sensing observations, as well as independent tracer-oceanographic estimates (Loose et al., 2009; Rodehacke et al., 2006). Just as regional atmospheric models will be key for evaluating the atmospheric component of climate models, regionally focused ocean models (e.g., Timmermann et al., 2012) and ocean reanalysis products (e.g. Menemenlis et al., 2008) are likely to provide valuable insight for evaluating CMIP ocean models.

### 3.2    Experiments with climate models coupled to ISMs

The second component of ISMIP6 is a suite of experiments designed to assess the impacts of dynamic ice sheets on climate and to better understand feedbacks between ice sheets and climate. We also aim to obtain an ensemble of sea-level projections from fully coupled atmosphere – ocean – ice sheet frameworks, which can later be compared to projections from standalone ice sheet models (Sect. 3.3). The experiments should be identical to the corresponding standard CMIP AOGCM experiments except for the treatment of ice sheets, so that any observed feedbacks and impacts can be attributed to dynamic ice sheets and not to other sources. As indicated in Table 2, four coupled AOGCM-ISM simulations are proposed, whose experiment IDs are *piControl-withism*, *1pctCO2to4x-withism*, *historical-withism* and *ssp585-withism*. These simulations are complemented by four ISM simulations: *ism-piControl-self, ism-1pctCO2to4x-self, ism-historical-self and ism-ssp585-self*.

In the *XXX-withism* setup, the ice sheet model is run interactively with the AOGCM: the climate model sends a surface forcing (SMB at a minimum) to the ice sheet model, and receives changes in ice sheet geometry. The land surface type and surface elevation in the climate model are dynamic, allowing, for example, a reduced albedo if the land surface changes from glaciated to unglaciated. Changes in the ice sheet mass should also affect the ocean temperature and salinity, as freshwater fluxes (liquid and/or solid) and energy fluxes are routed to the ocean. Liquid fluxes can originate from surface runoff, subglacial drainage systems, or basal melting of the ice in contact with the ocean. Solid fluxes come from iceberg calving, which may be computed with calving laws whose details are left to the discretion of the modeling groups. Explicit iceberg

models are not required. Similarly, ocean melting of ice shelves can be handled as desired, as long as the net freshwater flux and latent heat flux are routed consistently to the ocean model.

The *ism-XXX-self* configuration denotes runs of an uncoupled ice sheet model driven by the outputs of the AOGCM-only simulation (Sect. 3.1). The *ism-XXX-self* experiment is only meaningful in combination with a completed *XXX-withism*, and with the same combination of climate and ice sheet models. In this configuration, changes in the ice sheet do not affect the climate model, and therefore the climate inputs passed to the ice sheet model differ from those in the AOGCM-ISM experiment. The ice sheet model should, however, be configured with the same settings as for the AOGCM-ISM runs and should use the same initial conditions (i.e., the outcome of the spin-up carried out with the coupled AOGCM-ISM).

Initial conditions for both the *ism-XXX-self* experiments and the *XXX-withism* experiments will be generated by running the coupled AOGCM-ISM to a quasi-equilibrium state with pre-industrial forcing that represent year 1850. Pre-industrial AOGCM-ISM spin-up is an area of active research (e.g., Fyke et al. 2014) that seeks to produce a consistent non-drifting coupled state corresponding to the pre-industrial climate, which is different from the contemporary state (Kjeldsen et al., 2015). The challenge is that ice sheets reach quasi-equilibrium on timescales of many millennia, more slowly than the oceans, which typically have been the slowest components of AOGCMs. To reach steady state, the ice sheet model may have to be run for ~10,000 years or longer. Since runs of this length are impractical for a complex climate model, the coupling between the ice sheet model and the climate model will likely have to be asynchronous for at least part of the spin-up. In this case, once the ice sheet model has reached steady state, the coupled system should be run synchronously for an additional period before starting the experiments. ISMIP6 will not dictate spin-up procedures for obtaining pre-industrial initial ice-sheet conditions, but the procedure should be documented.

Ideally, the ice sheet model should be forced with the actual SMB computed by the climate model, rather than an SMB corrected to match observed climatology. We accept that there may be biases in the atmospheric or land models that can lead to an unrealistic SMB, which could result in a steady-state ice sheet geometry that differs substantially from present-day observations. However, correcting for these biases can distort the feedbacks between ice sheets and climate that we seek to investigate. We hope to learn from and ultimately reduce these biases, in the same way that biases elsewhere in the simulated coupled climate system are reduced by greater understanding and improved model design. On the other hand, if the geometry of the spun-up ice sheet is greatly different from observations, then the initial ice sheet for the *ism-XXX-self* experiments may be far from steady state with the SMB forcing from the standard, uncoupled AOGCM. As a result, the *ism-XXX-self* experiment could have a large drift that obscures the climate signal. The drift will be quantified from the control experiments. In case of a large drift, or if the spun-up ice sheet in the coupled system is deemed to be too unrealistic, an

alternative spin-up method would be to apply SMB anomalies from the AOGCM, superposed on a climatology that yields more realistic equilibrium ice sheet geometry.

The method used to downscale SMB (as well as oceanic forcing) from the coarse climate model grid to the finer ice sheet model grid is left to the discretion of each group, but should be well documented. The data request for ISMIP6 in Appendix A asks modelers to report certain fields on both the atmospheric and ice sheet grids to allow for an evaluation of the downscaling procedure. Also, ISMIP6 prefers that the surface-melt component of SMB be obtained from an energy-based method that conserves mass and energy, to facilitate interpretation of the drivers of SMB variability and change (e.g. Vizcaíno, 2014). Highly parameterized methods of computing surface melt, such as positive-degree-day (PDD) methods (e.g. Reeh, 1991; Bougamont et al., 2007), should be avoided. The choice of the ice sheet model, its complexity in approximating ice flow, and ice-sheet-relevant boundary conditions (e.g., geothermal flux) are left to the modelers' discretion. In all experiments, however, the ice sheets should not be forced to terminate at the present-day ice margin if the simulated SMB and/or the ice sheet dynamics cause a margin advance.

Regardless of the spin-up method, the first ISMIP6 experiment to be performed with the coupled AOGCM–ISM is the pre-industrial control, *piControl-withism*. This is a multi-century (500 years suggested) control run aiming to assess model drift and systematic bias and to capture unforced natural variability. The drift in the standalone ISM experiments *ism-XXX-self* will be quantified with a control run (*ism-piControl-self*). The core ISMIP6 prognostic climate change experiment is *1pctCO2to4x-withism*, which applies a 1% per year increase in $CO_2$ concentrations over 140 years until levels are quadrupled, then holds concentrations fixed for an additional two to four centuries. The *1pctCO2to4x-withism* will be compared to the AOGCM simulation *1pctCO2to4x* (the first 140 years of the DECK *1pctCO2* merged with the *1pctCO2-4xext*) and to *ism-1pctCO2to4x-self* (the standalone ISM forced by the AOGCM surface mass balance and temperature from *1pctCO2to4x*). The duration of these three experiments should be the same. It is suggested that the experiments be run for at least 350 years, and if possible for 500 years, because previous studies (e.g., Ridley et al., 2005; Vizcaíno et al., 2008; 2010) indicate that coupled AOGCM–ISM runs start to clearly diverge from uncoupled runs after about 250–300 years of simulation.

Another set of experiments repeats the CMIP6 *historical* and *ssp585* simulations with a coupled AOGCM-ISM. The *historical-withism* simulation begins at year 1850 from the pre-industrial spin-up and finishes at the end of 2014. This simulation is followed by *ssp585-withism,* with experimental settings and forcings as described in O'Neill et al. (2016). The *ssp585-withism* begins in January 2015 and is initiated from the December 2014 results of the *historical-withism* simulation. The *ssp585-withism* experiment is run for the 21[st] century and its extension to the end of the 23[rd] century. For completeness, these experiments are to be repeated with standalone ISM simulations *ism-historical-self* and *ism-ssp585-self*. We accept that

with this protocol, the 2015 ice sheet is likely to be distinct from the observed ice sheet due to model drift from the Historical run, and that this will have implications for projected ice sheet evolution (e.g., Stone et al., 2010).

Based on community feedback, we expect that several AOGCM–ISMs will be ready to participate in coupled climate experiments for CMIP6. Table 3 shows climate modeling centers that have expressed interest in participating in ISMIP6. The primary focus is coupled ice-sheet–atmosphere simulation for the Greenland Ice Sheet, but some groups have indicated participation only in the diagnostic aspect of ISMIP6 (where the goal is to provide climate data for the standalone ice sheet work). Full coupling of ice sheet models to climate models remains challenging, especially for interactions with the ocean. Accurate treatment of ice-ocean interactions requires ISMs that can simulate grounding line migration (which demands fine grid resolution) and iceberg calving, and ocean models that can simulate circulation in the cavities below ice shelves and the consequent melting or accretion of ice on the undersides of the shelves. Accurate treatment of ice-ocean interactions will likely also require ocean models to alter their domain (both vertically and horizontally) as the calving front migrates and as sub-ice-shelf ocean cavities evolve in space and time. For the Greenland Ice Sheet, ocean models may need to capture fjord dynamics on smaller spatial scales (~1 km) than are currently resolved by global ocean models. In addition, credible ice-ocean coupling requires accurate knowledge of the bathymetry beneath ice shelves and ice sheets, where data are sparse. Because of these challenges, we do not expect a realistic treatment of the Antarctic Ice Sheet in the ISMIP6 coupled AOGCM-ISM experiments. Antarctica is included, however, in the standalone experiments described in the next section.

### 3.3 Experiments with ISMs not coupled to climate models

The final set of ISMIP6 experiments will use standalone ice sheet models driven by climate model output and other datasets. Groups and models that have expressed an interest in participating in this aspect of ISMIP6 are listed in Table 4. The models participating in this effort will likely be configured differently from those in the *ism-XXX-self* simulations described in Sect. 3.2. For example, an ice sheet model that is spun up to quasi-equilibrium with a climate model will likely have a thickness and extent that differ appreciably from observed values, whereas standalone models can be initialized more realistically. Also, an ISM in a climate model might use a coarse resolution or a simple approximation of ice dynamics in order to be more computationally efficient, while the same model used strictly for projections would likely have a finer resolution, at least in regions of fast flow (e.g. Aschwanden et al., 2016), and could incorporate more complex ice flow dynamics. Similarly, ice sheet models that are used for paleoclimate studies are often distinct from those used for projections of a few hundred years.

### 3.3.1 initMIP

The initMIP ice sheet experiments are designed to explore uncertainties in sea-level projections associated with model initialization and spin-up. Such uncertainties have been identified by previous model intercomparison efforts (e.g.,

Bindschadler et al., 2012; Nowicki et al., 2013a, b; Edwards et al., 2014a, Shannon et al., 2013; Goelzer et al., 2013; Gillet-Chaulet et al., 2012) and include the impacts of model initial conditions, sub-grid scale processes, and poorly known parameters. The initMIP project aims to evaluate initialization procedures, to estimate trends caused by model initializations and to investigate the impact of choices in numerical and physical parameters (e.g., stress balance approximation or model resolution). Results of the initMIP project are expected to point to specific aspects of ice sheet initialization that have a crucial impact on sea-level projections and may be improved.

ISM initialization methods to present-day conditions range from running paleo-climate spin-up for thousands of years (e.g., Martin et al., 2011; Sato and Greve, 2012; Aschwanden et al., 2013; Fürst et al., 2015; Saito et al., 2016) to assimilating present-day observations (e.g., Morlighem et al., 2010; Gillet-Chaulet et al., 2012; Seroussi et al., 2013, Arthern et al., 2015). The choices made in this procedure affect ice sheet extent, flow rates, volume, and volume trends, which can have substantial effects on estimates of ice sheet contribution to sea-level rise (e.g. Aðalgeirsdóttir et al., 2014). Improving ISM initial conditions is an active area of research and a multidisciplinary effort. It requires acquisition of additional data with high spatial coverage over entire ice sheets and at increased resolution (e.g., Bamber et al., 2013; Rignot et al., 2011b; Joughin et al., 2010a; Howat et al., 2014). Ideally, all datasets used in the data assimilation are from the same period, as initializing an ice sheet model with datasets taken at different times can cause the ice flow model to artificially redistribute the glacier mass in unrealistic ways that serve only to reconcile these inconsistencies (Seroussi et al., 2011). This also implies that the date associated with the initial state can differ between models based on the data sets used. New algorithms that reconcile initialization datasets are being developed, most notably for bedrock elevation (e.g., Morlighem et al., 2011; Morlighem et al., 2014), which is notoriously poorly constrained.

The initMIP project consists of a Greenland component and an Antarctic component. Following initialization, there is a set of two forward experiments for the Greenland Ice Sheet and three forward experiments for the Antarctic Ice Sheet, each run for at least 100 years: i) a control run (*ism-ctrl-std*), ii) a surface mass balance anomaly run (*ism-asmb-std*) and iii) a basal melt anomaly run (*ism-abmb-std*) in which anomalous melt is applied beneath the floating portion of the Antarctic Ice Sheet. All other model parameters and forcing in the forward runs are the same as those used for initialization. The *ism-ctrl-std* is an unforced forward experiment designed to evaluate the initialization procedure and characterize model drift, the surface mass balance remaining identical to the one used during the initialization procedure. In *ism-asmb-std*, a prescribed SMB anomaly is applied to test the model response to a large perturbation. The schematic perturbation anomaly mimics outputs of several SMB models of different complexity between the end of the 20[th] century and the end of the 21[th] century, and is designed to capture the first-order pattern of SMB changes expected from climate models. The schematic SMB anomalies are defined everywhere on the model grid, and are therefore applicable for models with varying ice sheet extent. In *ism-abmb-std*, a prescribed anomaly of basal melting rate under floating ice is applied while SMB is kept the same as in *ism-ctrl-*

*std*. Because of the difference in ice shelf extent between the different models, the basal melt anomaly is prescribed to be constant for each basin. This scalar value is different for each basin and derived from the mean values of the ice shelf melt observed by Rignot et al. (2013) and Depoorter et al. (2013). The applied anomaly simulates a doubling of sub-ice-shelf melting after 40 years of simulation for models with initial melting rates close to today's observations.

Since these experiments are designed to allow comparison among the different models, some simplifications are imposed. Neither SMB nor bedrock topography should be adjusted in response to ice-sheet geometric changes in these forward experiments. However, to sample the uncertainty in sea-level due to initialization, groups are encouraged to submit multiple variations of the experiment, for example by changing the sliding law, stress balance approximation, model resolution, or

10 datasets (such as using different bedrocks). While the initialization procedures used by the different participating groups are not prescribed by ISMIP6, it is expected that individual groups will take advantage of the initMIP results to improve their initialization procedures. initMIP is also intended to give ice sheet modelers an opportunity to get involved in ISMIP6 at an early stage, before outputs of CMIP6 AOGCM become available; hence our prescription of simplified anomalies. We refer interested readers to the initMIP webpage ([http://www.climate-cryosphere.org/wiki/index.php?title=InitMIP](http://www.climate-cryosphere.org/wiki/index.php?title=InitMIP)) for more

information.

### 3.3.2     ism-XXX-std configuration

The *ism-XXX-std* experiments target primarily the glaciology community and seek to obtain realistic ice sheet evolution to inform estimates of past, present and future sea level. ISMIP6 will supply forcing data from CMIP6 that allows standalone ISMs to simulate the evolution of both the Greenland and Antarctic Ice Sheets. ISMIP6 seeks to assess the uncertainty in

sea-level change arising from both the ice sheet models and the climate forcing. A key concern is that ISMIP6 assess uncertainty associated with emission scenario and the AOGCMs' simulation of these scenarios: for a given emission scenario, the AOGCMs simulation of this scenario will result in a range of atmospheric and oceanic forcings. Clearly, there is a tension between the range of potential ice sheet forcing, the need to explore uncertainty associated purely with ISMs (e.g., related to initial conditions, bedrock topography and parametric uncertainty), and the computing requirements of

specific ISMs (some of which may only be able to perform a small number of experiments). To this end, we anticipate identifying a subset of forcing from the CMIP6 AOGCM ensemble based on the analysis of AOGCM simulations of ice-sheet climate (Sect. 3.1). The subset will be chosen to capture the full range of potential ice sheet forcing for a given emission scenario, using metrics of the SMB and ocean forcing to investigate that range. Within the selected subset of forcing, we plan to identify a small number of simulations that all ISMs must perform. Groups that are able to perform

numerous simulations will be encouraged to participate in all experiments. Shannon et al. (2013) is an example of this approach.

The forcing data can naturally be divided into atmospheric and oceanic forcing. Central to the former is the means to determine SMB associated with a particular CMIP6 experiment. Several methods have previously been employed to do this. Until we can assess the quality of the climate simulated by CMIP6 AOGCMs above and around the ice sheets (after the analysis of the CMIP6 DECK and Historical simulations), a definitive choice cannot be made. However, we list the options in order of preference:

1. Use the SMB calculated by the AOGCM directly. This has the advantage that the SMB will be entirely consistent with other parts of that AOGCM's simulation of climate. There is concern, however, that the quality of the SMB computed by the AOGCMs will make this approach unrealistic due primarily to the mismatch between the spatial resolution of AOGCMs and the characteristic length scale of variations in SMB. Several groups have, however, made recent progress in this area (e.g., Vizcaíno et al., 2013; Lipscomb et al., 2013). The use of anomalies should also be considered in this context.

2. In the event that AOGCM-determined SMB is shown to be inadequate, an intermediate step is required. Previously, this has been the use of Regional Climate Models (RCMs) to simulate SMB. For example, the ice2sea effort chose to generate SMB from an RCM (Edwards et al., 2014a,b; Fettweis et al., 2013). This approach, however, introduces a further link into the processing chain that may lead to delay in the production of sea-level projections. It also introduces the issue of choice of RCM and whether results from a number of RCMs should be used (further complicating the design of the ISM ensemble). Furthermore, the use of RCMs as intermediaries between AOGCMs and ISMs adds ambiguity about which biases are introduced by the AOGCMs and which biases are the result of the RCMs.

3. Use a parameterization or simplified process model to simulate SMB by downscaling atmospheric forcing over the ice sheet from an AOGCM. This approach was used by SeaRISE (Bindschadler et al., 2013), where the precipitation and surface temperature from 18 AOGCMs models taking part in the A1B scenario were combined to generate monthly mean values. These mean precipitation and temperature values were then passed to the SMB scheme of the ice sheet model (generally a PDD method that accounted for the temperature aspect of the SMB-elevation feedback) to obtain SMB anomalies that were added to the ice sheet surface conditions at initialization.

A further consideration is that the AOGCM models assume a fixed ice sheet elevation: i.e., they neglect the effect of ice sheet elevation change on the atmosphere and hence omit the SMB-elevation feedback. Standalone ISMs will need to include this effect by parameterizing the SMB lapse rate (Edwards et al., 2014a,b; Fettweis et al., 2013; Goelzer et al., 2013). This approach may be less of an issue for method 3 above because SMB is determined interactively within the ISM rather than being prescribed as forcing.

A second way in which the atmosphere could force dynamic change in ice sheets is through the production of large quantities of melt water. Mechanisms have been proposed that link melt water to both ice shelf collapse (Banwell et al.,

2013) and enhanced lubrication of ice flow (Zwally et al., 2002) (although recent modeling studies suggest a minor influence of the latter on large-scale ice flow (e.g., Shannon et al., 2013)). Surface air temperature and runoff forcing will therefore also be made available.

Both Antarctica and Greenland are thought to respond to changes in proximal ocean temperatures, which affect the melt rates of floating ice shelves and the vertical faces of outlet glaciers. Obtaining suitable oceanic forcing from CMIP6 climate models will be a major challenge. Few CMIP6 models will calculate the appropriate melt rates, and even these results are likely to be inaccurate because of issues of model resolution and the unique physics of ocean circulation adjacent to melting ice. Melt rates will therefore need to be determined outside the climate model using an index for proximal ocean
temperature. This index is most likely to be water temperature (and salinity) at the continental shelf break at an intermediate range of depths (equivalent to the base of ice shelves or the depth of ice grounded on bedrock). This quantity will be included in our evaluation of CMIP6 forcing (see Sect. 3.1).

A wide range of approaches has been used to calculate the required melt rate from prescribed ocean-temperature forcing. The
simplest method is to calculate melt rate anomalies from changes in the nearest ocean temperature using an observationally derived relation of 10 m yr$^{-1}$ °C$^{-1}$ (Rignot and Jacobs, 2002). However, this linear relation between ocean temperature and melt rates is calibrated for melt rates at the grounding line, and likely is missing important non-linearities (Holland et al., 2008). An alternative approach is to parameterize melt rates as proportional to the difference between ocean temperature at the shelf break and the freezing temperature at the ice shelf base. Beckman and Goosse (2003) developed such a scheme for
ocean models, and similar schemes have been applied in offline ice sheet model simulations with idealized ocean forcing (e.g., Martin et al., 2011; Pollard and DeConto, 2012; DeConto and Pollard, 2016). In those studies, the ocean temperature is set to the average temperature between 200 and 600 m depth (Martin et al., 2011), or the temperature at 400 m depth (DeConto and Pollard, 2016), or specified differently for specific Antarctic sectors (Pollard and DeConto, 2012). Depending on the evaluation of the CMIP6 models, ISMIP6 may adapt one of these choices, or could prescribe depth-varying profiles of
ocean temperature (and possibly salinity). The dependence of melt rates on thermal driving ranges from linear (Martin et al., 2011) to quadratic (Pollard and DeConto, 2012; DeConto and Pollard, 2016). Since the freezing temperature at the ice base decreases with depth, the melt rates in all schemes tend to be higher near grounding lines, as found from observations.

If none of the CMIP6 ocean models can accurately capture the broad-scale polar ocean circulation or produce realistic near-
shelf temperatures, an alternative is to prescribe a melt rate that simply depends on the ice shelf draft (e.g. Joughin et al., 2010b; Favier et al., 2014). This approach is less satisfactory, however, as it ignores temporal changes in ocean conditions, and typically uses coefficients calibrated to local thermal conditions. If ISMIP6 uses this approach, the provided coefficients would not be uniform, but would take into account that ocean waters reaching ice shelf cavities or fronts differ regionally. In

Antarctica, for example, the ice shelves of Pine Island Glacier and Thwaites Glaciers lie in "warm" water, while the Filchner-Ronne or Ross ice shelves reside in "cold" water. Ocean temperatures reflect the dominant water sources, with warm waters dominated by circumpolar deep waters (Jacobs et al., 2011), while cold waters typically correspond to high-salinity shelf water (Nichols et al., 2001).

Ice-ocean interactions are an active area of research, and more sophisticated parameterizations of melt are becoming available (e.g., Jenkins, 2016; Asay-Davis et al., 2016). Simplified models of the system could be used (e.g., Payne et al., 2007), as could high-resolution ocean models that resolve ice-shelf cavities and fjords. Given this wide range of methods, ISMIP6 will leave the detailed choice of the parameterization to individual ice-sheet modelers, but will issue guidance on

10 what constitutes an acceptable parameterization. We will organize workshops with the polar ocean community to investigate how to best derive oceanic forcing for ice sheet models, so that by the time the CMIP6 ocean models are evaluated, a clearer protocol is in place. The calculated melt rate will be part of the standard data request for ice sheet models (see Appendix A), and part of our evaluation will be to determine how well the applied forcing compares to observed melt rates of Rignot et al. (2014) and Depoorter et al. (2014).

ISMIP6 will not dictate the choice of ice sheet model complexity in terms of the ice flow approximation, the basal sliding law, the treatment of grounding lines, the calving law, the ice-sheet-specific boundary conditions (e.g., bedrock topography), or the initialization method. An exception is that models of the Antarctic Ice Sheet should include floating ice shelves and grounding line migration. The spatial resolution of the ISM in the vicinity of fast-flowing ice streams and the grounding line

affects the dynamic response (Durand et al., 2009; Pattyn et al., 2012, 2013), and the model resolution must be fine enough to capture this response accurately. To this end, participating models are encouraged to take part in model intercomparison efforts that target specific aspects of ice sheet modeling, such as the current MISOMIP (Marine Ice Sheet–Ocean Model Intercomparison Project; Asay-Davis et al., 2016) and are required to take part in initMIP (initialization-focused experiments that compare and evaluate the simulated present-day state; Sect. 3.3.1). The lack of a stricter protocol is a reflection of the

challenges in identifying which factors are the most important when making projections, which datasets are most accurate, and how to best capture and parameterize certain ice-sheet processes. For example, although the choice of bedrock topography affects mass transport and is thus likely to influence a projection, it is currently not possible to identify a best dataset due to the difficulty in obtaining bedrock measurements. Groups are encouraged to repeat the experiments with a variety of perturbations of weakly constrained parameters, boundary conditions, etc. in order to test the sensitivity of

projections to these choices.

Unlike the protocol for climate models, the *ism-XXX-std* simulations cannot be initiated from a spin-up corresponding to year 1850. This is due to the challenge of initializing ice sheet models to pre-industrial conditions, which are constrained more

weakly than the contemporary state: the quantity of accurate, high-resolution data available during the satellite era far exceeds that available for pre-industrial and historical periods. The majority of ice sheet models use these data in sophisticated initialization and assimilation procedures, such that the present-day state of the ice sheet is simulated with high fidelity. The lack of suitable data before the satellite era means that no such accuracy can be assumed for simulations of the historical periods. Such inaccuracies are known to have a large effect on projections. For instance, discrepancies between projections can often be attributed to slight differences in the geometry (e.g., Shannon et al. 2013). The *ism-XXX-std* simulations will thus be initiated from a present-day spin-up.

The first *ism-XXX-std* simulation is *ism-pdControl-std*, the ice sheet present-day control with constant forcing needed to evaluate model drift. This constant forcing is based on the climate at the end of the initialization procedure. For many models, the forcing and simulation will be the same as the *ism-ctrl-std* in the initMIP experiment (Sect. 3.3.1), unless a change has been made in the initialization. The idealized climate change experiment, *ism-1pctCO2to4x-std*, considers a 1% per year atmospheric $CO_2$ concentration rise until quadrupled concentrations and stabilization thereafter. The *ism-historical-std* will be an abbreviated simulation for the historical period (as it begins from the present-day spin-up) and, following the CMIP6 protocol, ends in December 2014. The *ism-amip-std* is a simulation for the last few decades to understand the well-observed record of ice sheet changes. The results from the *ism-amip-std* and *ism-historical-std* are likely to differ, and the comparison will provide some insight into the relative importance of biases, climate variability and climate change. The main simulation for projecting 21st century sea-level rise is the *ism-ssp585-std*, which is initiated from the *ism-historical-std* simulation. (As mentioned previously, other scenarios will be considered if time permits.) If possible, projections should continue to the end of the 23rd century.

We complement the experiments for the recent past and future with one paleo experiment (*ism-lig127k-std*), to simulate Greenland ice-sheet evolution during the Last Interglacial. The transient simulation will span the period 135 kyr to 115 kyr to include transitions from the preceding and to the following cold periods. The climate forcing for *ism-lig127k-std* will be derived from the PMIP4-CMIP6 experiment *lig127k* and other (transient) LIG climate simulations (cf. Bakker et al., 2012; Lunt et al., 2013) that will be performed by PMIP4 (Otto-Bliesner et al., 2016). The proposed experiment builds on past efforts to study Greenland ice-sheet stability and evolution during the LIG and constrain the Greenland contribution to the LIG sea-level highstand (e.g. Robinson et al., 2011; Born and Nisancioglu, 2012; Helsen et al., 2013).

### 3.4    Prioritization of experiments and timing

The ISMIP6 experiments listed in Table 2 are divided into three "Tiers" to indicate prioritization. Tier 1 denotes experiments that are to be completed by the ISMIP6 participants. Tier 2 experiments are highly encouraged, while Tier 3 experiments are optional.

For the coupled AOGCM-ISM experiments, the Tier 1 experiments *piControl-withism* and *1pctCO2to4x-withism* should be performed first. These experiments have already been performed by many climate modeling groups, and their idealized settings allow for an easier evaluation of the ice-climate feedback. The Tier 2 experiments, *historical-withism and ssp585-withism*, are more relevant to our goal of producing sea-level projections concurrent with the CMIP6 future climate. Ideally, the *XXX-withism* and *ism-XXX-self* experiments would follow the corresponding AOGCM experiments with no more than a six-month lag.

For the standalone *ism-XXX-std* experiments, ISMIP6 is constrained by the timing of the AOGCM runs that will be used to derive forcings for ice sheets. We anticipate that the DECK simulations will not be completed before spring of 2017, which implies that climate models cannot be evaluated rigorously before summer 2017, and in turn that the ISM Tier 1 experiments based on CMIP6 DECK forcing would begin in 2018. As soon as suitable forcings are available from the SSP5-8.5 experiments (CMIP6-Endorsed ScenarioMIP, Tier 1), *ism-ssp585-std* will be the focus of the standalone ISM work. To allow ice-sheet modeling groups the necessary time to perform the simulations, we plan to begin *ism-ssp585-std* in early 2019. Similarly, the *ism-lig127k-std* cannot proceed until the PMIP participants have completed the CMIP6-Endorsed PMIP4 Tier 1 experiment and other transient PMIP4 experiments. In the meantime, ISMIP6 standalone ice sheet models will focus on initMIP, with the goal of finishing this suite of experiments by the end of 2016 for Greenland and by the end of 2017 for Antarctica.

## 4  Evaluation and Analysis

The framework described in this paper entails an evaluation of the climate system, with a particular focus on the polar regions. This framework works toward the goals of i) assessing the effect of including dynamic ice sheets in climate models and ii) improving confidence in projections of sea-level rise associated with mass loss from the Greenland and Antarctic Ice Sheets. Our evaluation and analysis will be based on key model output variables for the atmosphere, ocean and ice sheets that form the ISMIP6 data request summarized in Appendix A.

### 4.1  Evaluation of ice sheet models

Ice sheet models will be evaluated using methodologies already in use by the ice-sheet modeling community. These metrics typically begin by assessing whether the volume and area of the modeled present-day ice sheet are comparable to observed values. The next step evaluates the spatial patterns of surface elevation, ice sheet thickness, surface velocities, and positions of the ice front and grounding line. Some ice sheet models are initialized using data assimilation methods, which precludes the use of certain observations in the evaluation. Evaluation of these models can be done by hindcasting, a method that evaluates whether recent observed trends are captured (Aschwanden et al., 2013). Examples include comparison against the

gravimetry (GRACE) time series from 2003 onwards, which provides an integrated set of measurements for mass changes in Greenland and Antarctica. This approach will also enable a direct comparison between predicted sea-level rise from ISMs and the change in ocean mass observed by GRACE. The recent IMBIE effort (Ice Sheet Mass Balance Inter-comparison Exercise, Shepherd et al., 2012) facilitates this comparison by combining observations from gravimetry, altimetry and velocity changes between 1992-2012 into a single dataset of annual mass budget for each ice sheet. The follow-on effort, IMBIE2 (Shepherd, personal communication), will extend the record in time and plans to separate the observed mass change into SMB and dynamic components.

## 4.2 Effects of dynamic ice sheets on climate

The combination of coupled AOGCM-ISM simulations (*XXX-withism*) and standalone ice sheet simulations (*ism-XXX-self*) will support a clean analysis of ice-sheet feedbacks on the climate system, which can further affect ice-sheet evolution (e.g., Driesschaert et al., 2007; Goelzer et al., 2011; Vizcaíno et al., 2008, 2010, 2015). A limited number of feedbacks can be studied in an AOGCM without a dynamic ISM. For instance, because AOGCMs generally compute ice-sheet SMB through a land model coupled on hourly time scales to the atmospheric model, the albedo-melt feedback can be studied in an AOGCM alone. Other important feedbacks, however, are present only if the ice sheet is dynamic:

- As ice sheets thin, the lower elevation leads to warmer surface temperatures that increase melting. This ice-elevation feedback is small on sub-century time scales (Edwards et al., 2014b), but over longer time scales, it can drive ice sheets to a point of no return, where retreat would continue unabated even if the climate returned to an unperturbed state.
- Changes in ice sheet elevation modify the regional atmospheric circulation (e.g., Ridley et al., 2005), which can either enhance or slow the rate of retreat.
- Changes in land surface cover (e.g., from glaciated to vegetated) can darken and warm the surface, promoting atmospheric warming and further melting.
- Increased freshwater fluxes (both solid and liquid) from retreating ice sheets can modify the density structure of the ocean, possibly suppressing convection and weakening the Atlantic meridional overturning circulation. Although some studies (e.g., Hu et al., 2009) find that this is a small effect, others suggest that increased runoff from the Greenland Ice Sheet has already reduced deep convection in the Labrador Sea (Yang et al., 2016).
- The buoyancy of fresh glacial meltwater from sub-ice-shelf melting can modify the ocean circulation that drives the melting. On longer time scales, changes in the size and shape of sub-shelf cavities may also alter the circulation.

The ISMIP6 experiments will be performed on climate model runs lasting several centuries, long enough to allow a detailed analysis of at least the first four of these feedbacks. Ocean cavity feedbacks, however, may require further development of ocean models that can adjust their boundaries dynamically as marine ice sheets advance and retreat.

### 4.3    Sea-level change

The SMB over the Greenland Ice Sheet is currently becoming less positive, thus resulting in an increasing contribution to sea-level rise due to increased surface runoff (van Angelen et al., 2014; Fettweis et al., 2011). This trend is expected to continue (Fettweis et al., 2013; Rae et al., 2012), although there is a large spread in AOGCMs (Yoshimori and Abe-Ouchi, 2012). The picture is less clear for the Antarctic Ice Sheet, where both accumulation and surface melt are projected to increase (Lenaerts et al., 2016). The multi-model ensemble of the surface freshwater flux from AOGCM simulation will provide insight into the resulting contribution of past and future sea level due to changes in SMB alone.

The largest uncertainty in sea level, however, remains the dynamic contribution from the ice sheets. ISMIP6 targets the contribution of dynamic ice sheets to global sea level, via multi-model ensemble analysis of standalone ice sheet models (*ism-XXX-std*). For a number of experiments, the multi-model ensemble from the *ism-XXX-std* will be contrasted to the multi-model ensemble resulting from coupled AOGCM-ISM simulations (*ism-XXX-withism*). We expect the results of the standalone modeling (*ism-ssp585-std*) to be more robust for projections, as we anticipate that the spun-up ice sheet from the coupled historical simulation (*historical-withism*) will differ substantially from present-day observations, and these differences will alter the projected ice sheet evolution (e.g., Stone et al., 2010; Shannon et al., 2013). The projections from *ssp585-withism* will likely expose issues resulting from coupling dynamic ice sheet models to climate models, motivating the community to begin resolving them.

We also aim to quantify the uncertainty in sea level arising from uncertainties in both the ice sheet models and the climate input, hence the need to sample across scenarios and models. For example, the ongoing initMIP project will provide insight into sea-level uncertainties resulting from ice sheet model initialization. By repeating model runs with different datasets, sliding laws, model resolutions, etc., initMIP will allow us to constrain the sea-level contribution associated with these choices. Ice sheet evolution will also depend on climatic drivers. For instance, given a certain number of AOGCMs that simulate present-day ice-sheet SMB reasonably well, comparing their SMB results under various climate-change simulations will allow us to quantify climate-model-driven uncertainty in SMB. If relationships between large-scale climate drivers (e.g., regional temperature and precipitation) and ice-sheet area-integral SMB can be established (e.g., Gregory and Huybrechts, 2006; Fettweis et al., 2013), this would allow estimation of SMB from AOGCM experiments for other climate scenarios. If possible, synergies with other CMIP6 efforts will allow us to further investigate the uncertainty in climate input. For example, the CMIP6-Endorsed High Resolution Model Intercomparison Project (HighResMIP, Haarsma et al., 2016) and Coordinated Regional Climate Downscaling Experiment (CORDEX, Gutowski et al., 2016) may allow us to quantify the impacts of increased resolution on SMB.

## 5   Discussion and conclusion

ISMIP6 has an experimental protocol and a diagnostic protocol. The experimental design uses and builds upon the core DECK and CMIP6 Historical simulations, along with selected CMIP6-Endorsed PMIP4 and ScenarioMIP simulations. The suite of ISMIP6 experiments involves three types of models: AOGCM/AGCM with no dynamic ice sheets, coupled AOGCM-ISM, and standalone ISM. The diagnostic protocol is based on ice-sheet-related model outputs, many of which are already present in the CMIP6 atmosphere and ocean diagnostics. The evaluation of the climate in the polar regions from AOGCM and AOGCM-ISM simulations will guide recommendations for existing and new ice-sheet–climate coupling efforts. ISMIP6 promotes the development of the ice sheet component of climate models in an effort to bring both climate and ice-sheet models to greater maturity. ISMIP6 targets two of the WCRP Grand Science Challenges: "Melting Ice and Global Consequences" and "Regional Sea-level Change and Coastal Impacts". Given the current rapid changes in the Greenland and Antarctic Ice Sheets, ice sheets no longer be considered passive players in the climate system. Their contributions to future sea level will likely have considerable human and environmental impacts, and ISMIP6 will facilitate research in this critical area.

ISMIP6 will coordinate simulation and analysis of ice sheet evolution in a changing climate. Inclusion of ice sheet models is unique in CMIP history and is necessary to advance understanding of the sea-level contribution from the Greenland and Antarctic Ice Sheets, the climate system response to ice-sheet changes, and the feedbacks between ice sheets and climate. ISMIP6 is thus an important step in closing the gap between the climate and ice-sheet modeling communities. Our key output, the sea-level contribution from ice sheets, complements the projections of ocean thermal expansion that already sit within the CMIP framework. This improvement will help sea level join the family of variables for which CMIP can provide routine IPCC-style projections. Ultimately, the success of ISMIP6 relies on the broad participation of the CMIP6 modeling centers, standalone ice sheet modeling groups, and analysts of the atmosphere, ocean and ice sheets.

### Data availability

The model output from the simulations described in this paper will be distributed through the Earth System Grid Federation (ESGF) with digital object identifiers (DOIs) assigned. In order to document CMIP6's scientific impact and enable ongoing support of CMIP, users are obligated to acknowledge CMIP6, the participating modeling groups, and the ESGF centers (see details on the CMIP Panel website at http://www.wcrp-climate.org/index.php/wgcm-cmip/about-cmip). Datasets for natural and anthropogenic forcings are required to run the experiments; these datasets are described in separate invited contributions to this Special Issue. The forcing datasets will be made available through the ESGF with version control and DOIs assigned. Exceptions in the distribution method will be made for the forcing for the initMIP Greenland and Antarctic efforts that specifically target standalone ice sheet models. Instruction of how to obtain forcing datasets not available through ESGF will be posted on the ISMIP6 website (http://www.climate-cryosphere.org/activities/targeted/ismip6).

**Acknowledgements**. We thank the CMIP6 panel members for their continuous leadership of the CMIP6 effort, the Working Group on Coupled Modeling (WGCM) Infrastructure Panel (WIP) for overseeing the CMIP6 and ISMIP6 infrastructure, and in particular Martin Juckes and Alison Pamment for their help with the ISMIP6 data request, and Karl Taylor for sharing his wisdom on CMIP experiment protocols. We thank the current ISMIP6 members, the modeling groups and the wider glaciology community for their contribution in the ISMIP6 design. We acknowledge the Climate and Cryosphere (CliC) Project and the World Climate Research Programme (WCRP) for their guidance, support and sponsorship. HG has received funding from the program of the Netherlands Earth System Science Centre (NESSC), financially supported by the Dutch Ministry of Education, Culture and Science (OCW) under Grant nr. 024.002.001. SN, HS and EL were supported by grants from NASA Cryospheric Science Program and the NASA Modeling Analysis and Prediction Program. WL was supported by the Regional and Global Climate Modeling program of the Office of Biological and Environmental Research within the US Department of Energy's Office of Science. WJP is supported by the NERC Centre for Polar Observation and Modelling (CPOM). We thank our topical editor Philippe Huybrechts, our reviewers Christian Rodehacke and Xylar Asay-Davis, and everyone who contributed to the open discussion for constructive comments.

The article processing charges for this open-access publication were covered by the NASA Cryosphere Program and the NASA Modeling Analysis and Prediction Program.

## 6 Appendix A: Variable Request

This special issue includes a manuscript that is dedicated to the CMIP6 data request. The majority of our data request is based on the CMIP5 CMOR tables Amon (Monthly Mean Atmopsheric Fields), Omon (Monthly Mean Ocean Fields), LImon (Monthly Mean Land Cryosphere Fields), and OImon (Monthly Mean Ocean Cryosphere Fields), which already contained many of the output required to diagnose and intercompare the climate over land ice/ice sheets and to derive forcing for the ice sheets. In the CF convention, 'land ice' comprises grounded ice sheets, floating ice shelves, glaciers and ice caps, while 'ice sheet' refers to grounded ice sheets and floating ice shelves. A few additional variables are needed to properly derive the forcings for ice sheets from AOGCMs, and to record outputs from the evolving ice sheets in the coupled AOGCM-ISMs experiments (such as ice elevation change), or from the standalone ice sheet simulations. In this Appendix, we briefly outline the ISMIP6 data request on the atmosphere grid (Table A1), ocean grid (Table A2), and ice sheet grid (Table A3), and provide some context for key new variables.

The mass change of ice sheets (see Fig. A1) is a result of the surface mass balance (SMB), ice melt (or refreeze) at the base of the grounded ice sheet (BMB), and mass exchange with the ocean. The latter can be further split into frontal mass balance (FMB, defined as iceberg calving and melt (or refreeze) at the ice shelf front) and melt (or refreeze) at the base of ice shelves

(BMB). All fluxes are defined as positive when the process adds mass to the ice sheet and negative otherwise. The thermal state of the ice sheet models is documented by the basal temperature and by the temperature at the ice sheet-snowpack interface. Note that BMB and basal temperature are computed differently depending on whether the ice is grounded or floating, requiring the use of distinct Long Names, but same Standard Names in Table A3.

Climate models will be evaluated primarily based on how well they can simulate SMB over the ice sheets. This quantity (see Vizcaíno (2014) and Fig. A2) can be defined as precipitation minus runoff minus evaporation (which in our context includes any sublimation, a small term over ice sheets), where precipitation is the sum of snowfall and rainfall. Runoff is the liquid water that escapes the ice sheet, while some of the water may be retained in the snow pack and possibly refreezes. The

10 evaluation of climate models also benefits from analysis of energy fluxes, key temperatures, and area fraction of land ice, grounded ice sheet (excludes ice shelf) and snow over the land ice. Note that some variables, such as SMB, are present in both Table A1 and Table A3, since in a coupled AOGCM-ISM simulation, the two will differ due to downscaling to the ice sheet grid. The data request for the ocean serves primarily as input to construct oceanic forcing for ice sheet models off-line. It is not as extensive as the data request for the atmosphere, because marine boundary conditions for outlet glaciers and ice

shelves are not routinely generated by AOGCMs. It is therefore premature to set diagnostic protocols at this stage. However, participants are asked to follow the protocols of the CMIP6-Endorsed Ocean Model Intercomparison Project (OMIP, Griffies et al., 2016) when preparing the data listed in Table A2, in particular when regridding the ocean data from a native grid to the CMIP6 standard grids. The ice sheet data request contains key characteristics needed to evaluate the ice sheet geometry, and ice sheet flow. It also contains key ice sheet specific boundary conditions that may differ between

models and a record of the forcing applied to the ice sheet model. To facilitate the analysis of the ice sheet contribution to sea level, a number of integrated measures (for example, ice sheet mass) are also requested.

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

## 8    Tables and Figures

Table 1: Overview of the experiments with climate models not coupled with ice sheet models that are to be used by ISMIP6. All experiments are started on 1 January and end on 31 December of the specified years.

| Experiment | CMIP6 label (experiment_id) | Experiment description | Start year | End year | Minimum no. years per simulation | Major purposes |
|---|---|---|---|---|---|---|
| **DECK experiments** | | | | | | |
| AMIP | *amip* | Observed SSTs and SICs prescribed | 1979 | 2014 | 36 | Evaluation, variability |
| Pre-industrial control | *piControl* | Coupled atmosphere-ocean pre-industrial control | n/a | n/a | 500 | Evaluation, unforced variability |
| 1% yr$^{-1}$ CO$_2$ concentration increase | *1pctCO2* | CO$_2$ concentration prescribed to increase at 1% yr$^{-1}$ | n/a | n/a | 150 | Climate sensitivity, feedbacks, idealized benchmark |
| **Extension of *1pctCO2*, needed to generate 1pctCO2to4x** | | | | | | |
| Extension from year 140 of *1pctCO$_2$* with 4 time CO$_2$ | *1pctCO2-4xext* | Branched from *1pctCO$_2$* run at year 140 and run with CO2 fixed at quadruple pre-industrial concentration | n/a | n/a | 210 | Climate sensitivity, feedbacks, idealized benchmark |
| **CMIP6 historical simulation** | | | | | | |
| Past ~ 1.5 centuries | *historical* | Simulation of the recent past | 1850 | 2014 | 165 | Evaluation |
| **CMIP6-Endorsed ScenarioMIP simulations** | | | | | | |
| SSP5-8.5 | *ssp585* | Future scenario with high radiative forcing by the end of the century | 2015 | 2100 | 86 | Climate sensitivity |
| SSP5-8.5ext | *ssp585* | Extension of high radiative forcing future scenario | 2101 | 2300 | 200 | Climate sensitivity |
| **CMIP6-Endorsed PMIP4 simulation** | | | | | | |
| PMIP4 last interglacial | *lig127k* | Equilibrium simulation of the peak of the last interglacial period | 127ka | n/a | 100 | Climate sensitivity, feedbacks, long responses |

Table 2: Overview of the ISMIP6 experiments with dynamic ice sheets that are either coupled to climate models (AOGCM-ISM, *XXX-withism*) or run offline (ISM, *ism-XXX-self* and *ism-XXX-std*). All experiments are started on 1 January and end on 31 December of the specified years. PD indicates that the start date correspond to the date of the present-day ISM spin-up.

| Experiment | CMIP6 label (experiment_id) | Experiment description | Start year | End year | Minimum no. years per simulation | Starting conditions | Tier |
|---|---|---|---|---|---|---|---|
| **Repeat of DECK experiments with dynamic ice sheets** | | | | | | | |
| AMIP | *ism-amip-std* | Offline ISM forced by ISMIP6-specified AGCM *amip* output | PD | 2014 | n/a | ISM spinup | 2 |
| Pre-industrial control | *piControl-withism* | Pre-industrial control with interactive ice sheet | n/a | n/a | 500 | AOGCM-ISM spinup | 1 |
| | *ism-piControl-self* | Offline ISM forced by own AOGCM *piControl* output | n/a | n/a | 500 | AOGCM-ISM spinup | 1 |
| Present-day control | *ism-pdControl-std* | Offline ISM forced by end of present-day spinup conditions | n/a | n/a | 100 | ISM spinup | 1 |
| **Repeat of 1pctCO2to4x with dynamic ice sheets** | | | | | | | |
| 1% yr$^{-1}$ $CO_2$ concentration increase to 4 time $CO_2$ | *1pctCO2to4x-withism* | Simulation with interactive ice sheet forced by 1% yr$^{-1}$ $CO_2$ increase to 4x $CO_2$ (subsequently held constant to quadruple levels) | n/a | n/a | 350 | AOGCM-ISM spinup | 1 |
| | *ism-1pctCO2to4x-self* | Offline ISM forced by own AOGCM *1pctCO2to4x* output | n/a | n/a | 350 | AOGCM-ISM spinup | 1 |
| | *ism-1pctCO2to4x-std* | Offline ISM forced by ISMIP6-specified AOGCM *1pctCO2to4x* output | n/a | n/a | 350 | ISM spinup | 1 |
| **Repeat of CMIP6 historical simulation with dynamic ice sheets** | | | | | | | |
| Past ~ 1.5 centuries | *historical-withism* | Historical simulation with interactive ice sheets | 1850 | 2014 | 165 | AOGCM-ISM spinup | 2 |
| | *ism-historical-self* | Offline ISM forced by own AOGCM *historical* output | 1850 | 2014 | 165 | AOGCM-ISM spinup | 2 |
| | *ism-historial-std* | Offline ISM forced by ISMIP6-specified AOGCM *historical* output | PD | 2014 | n/a | ISM spinup | 2 |
| **Repeat of CMIP6-Endorsed ScenarioMIP simulations with dynamic ice sheets** | | | | | | | |
| High radiative forcing future emission scenario (SSP5-8.5) | *ssp585-withism* | SSP5-8.5 simulation with interactive ice sheet | 2015 | 2100 | 86 | *historical-withism* | 2 |
| | *ism-ssp585-self* | Offline ISM forced by own AOGCM *ssp585* output | 2015 | 2100 | 86 | *ism-historical-self* | 2 |
| | *ism-ssp585-std* | Offline ISM forced by ISMIP6-specified AOGCM *ssp585* output | 2015 | 2100 | 86 | *ism-historical-std* | 2 |
| Extension of high radiative forcing future scenario (SSP5-8.5ext) | *ssp585-withism* | Extension of SSP5-8.5 simulation with interactive ice sheet | 2101 | 2300 | 200 | *ssp585-withism* | 3 |
| | *ism-ssp585-self* | Offline ISM forced by own AOGCM *ssp585* output | 2101 | 2300 | 200 | *ism-ssp585-self* | 3 |
| | *ism-ssp585-std* | Offline ISM forced by ISMIP6-specified AOGCM *ssp585* output | 2101 | 2300 | 200 | *ism-ssp585-std* | 3 |
| **Last interglacial simulation based on PMIP4 simulations with standalone ice sheet only** | | | | | | | |
| Last interglacial | *ism-lig127k-std* | Last interglacial simulation forced by *lig127k* and other PMIP experiments. | 135ka | 115ka | 20000 | ISM spinup | 3 |
| **initMIP Greenland and Antarctic simulations with standalone ice sheet only** | | | | | | | |
| Present-day control | *ism-ctrl-std* | Present-day control | n/a | n/a | 100 | ISM spinup | 1 |
| Surface mass balance | *ism-asmb-std* | Surface mass balance anomaly prescribed | n/a | n/a | 100 | ISM spinup | 1 |
| Basal melt | *ism-bsmb-std* | Basal melt anomaly under floating ice prescribed (Antarctica only) | n/a | n/a | 100 | ISM spinup | 1 |

Table 3: Climate Modeling Centers that have expressed an interest in ISMIP6. *Indicates only an interest in the diagnostic component (no AOGCM-ISM participation anticipated).

| Climate Model | Ice Sheet Model | Institute/Country |
|---|---|---|
| CanESM* | None | CCCma/CA |
| CESM2 | CISM | NCAR-LANL/USA |
| CNRM-CM | GRISLI | CNRM-CERFACS/FR |
| EC-Earth | GrIS | DMI/DK |
| GISS | PISM | NASA-GISS/USA |
| INMCM | VUB | INM/RU |
| IPSL-CM6 | GRISLI | IPSL/FR |
| MIROC-ESM | IcIES | AORI-UT-JAMSTEC-NIES/JP |
| MPI-ESM | PISM | MPI/DE |
| UKESM | BISICLES | MetOffice/UK |

Table 4: Ice sheet modeling groups that have expressed an interest in ISMIP6.
x Indicates planned contribution.

| Ice Sheet Model | Greenland | Antarctica | Institute/Country |
|---|---|---|---|
| BISICLES | x | | BGC/UK |
| CISM | x | | LANL/NCAR/USA |
| Elmer/Ice | x | x | LGGE/FR |
| f.ETISH | x | x | ULB/BE |
| GISM | x | | VUB/BE |
| GRISLI | x | | LSCE/FR |
| IcIES | x | x | MIROC/JP |
| IMAUICE | x | x | IMAU/NL |
| ISSM | x | x | JPL/USA |
| ISSM | x | x | UCI/USA |
| ISSM | x | | AWI/DE |
| MPAS-LI | | x | LANL/ORNL/USA |
| PennState3D | | x | PSU/USA |
| PISM | x | | UAF/USA |
| PISM | x | x | ARC/NZ |
| PISM | x | | DMI/DK |
| PISM | x | | MPIM/DE |
| SICOPOLIS | x | x | ILTS/JP |
| SICOPOLIS | x | x | PIK/DE |
| Úa | | x | BAS/UK |
| WAVI | | x | BAS/UK |

Table A1: Data in the LImon Table (Monthly Mean Land Cryosphere Fields) and/or Amon Table (Monthly Mean Atmospheric Fields) needed to capture the glaciated/ice sheet surface realm. These fields are saved on the atmosphere grid and contain monthly output. Tier indicate priority of variable: Mandatory (1), Desirable (2), Experimental (3). These variables are requested for climate models participating in the diagnostic component of ISMIP6 (Table 1), and for the *XXX-withism* experiments (Table 2). Flux variables are defined positive when the process adds mass or energy to the ice sheet and negative otherwise.

| Long name (netCDF) | Units | Standard name (CF) | Tier |
|---|---|---|---|
| Near surface air temperature (2m) | K | air_temperature | 1 |
| Surface temperature | K | surface_temperature | 1 |
| Snow internal temperature | K | temperature_in_surface_snow | 2 |
| Temperature at the top of ice sheet model | K | temperature_at_top_of_ice_sheet_model | 2 |
| Surface mass balance flux | $kg\,m^{-2}\,s^{-1}$ | land_ice_surface_specific_mass_balance_flux | 2 |
| Snowfall flux | $kg\,m^{-2}\,s^{-1}$ | snowfall_flux | 1 |
| Rainfall flux | $kg\,m^{-2}\,s^{-1}$ | rainfall_flux | 2 |
| Surface snow and ice sublimation flux | $kg\,m^{-2}\,s^{-1}$ | surface_snow_and_ice_sublimation_flux | 2 |
| Surface snow and ice melt flux | $kg\,m^{-2}\,s^{-1}$ | surface_snow_and_ice_melt_flux | 2 |
| Surface snow melt flux | $kg\,m^{-2}\,s^{-1}$ | surface_snow_melt_flux | 3 |
| Surface ice melt flux | $kg\,m^{-2}\,s^{-1}$ | land_ice_surface_melt_flux | 3 |
| Surface snow and ice refreezing flux | $kg\,m^{-2}\,s^{-1}$ | surface_snow_and_ice_refreezing_flux | 3 |
| Land ice runoff | $kg\,m^{-2}\,s^{-1}$ | land_ice_runoff_flux | 2 |
| Snow area fraction | 1 | surface_snow_area_fraction | 1 |
| Land ice area fraction | 1 | land_ice_area_fraction | 1 |
| Grounded ice sheet area fraction | 1 | grounded_ice_sheet_area_fraction | 1 |
| Floating ice shelf area fraction | 1 | floating_ice_shelf_area_fraction | 1 |

| | | | |
|---|---|---|---|
| Land ice altitude | m | surface_altitude | 1 |
| Net latent heat flux over land ice | W m$^{-2}$ | surface_upward_latent_heat_flux | 1 |
| Sensible heat flux over land ice | W m$^{-2}$ | surface_upward_sensible_heat_flux | 1 |
| Downwelling shortwave | W m$^{-2}$ | surface_downwelling_shortwave_flux_in_air | 1 |
| Upward shortwave over land ice | W m$^{-2}$ | surface_upwelling_shortwave_flux_in_air | 1 |
| Downwelling longwave | W m$^{-2}$ | surface_downwelling_longwave_flux_in_air | 1 |
| Upward longwave over land ice | W m$^{-2}$ | surface_upwelling_longwave_flux_in_air | 1 |
| Albedo over land ice | 1 | surface_albedo | 2 |

Table A2: Data on the Omon Tables (Monthly Mean Ocean Fields) needed to capture the glaciated/ice sheet surface realm or for intercomparison of the model simulations. These fields are saved on the ocean grid and contain monthly output. Data preparation should follow the CMIP6-Endorsed OMIP protocol. Tier indicates priority of variable: Mandatory (1), Desirable (2), Experimental (3). These variables are requested for climate models participating in the diagnostic component of ISMIP6 (Table 1), and for the *XXX-withism* experiments (Table 2). Flux variables are defined positive when the process adds mass to the ocean and negative otherwise.

| Long name (netCDF) | Units | Standard name (CF) | Tier |
| --- | --- | --- | --- |
| Global surface height above geoid | m | sea_surface_height_above_geoid | 1 |
| Global average thermosteric sea-level change | m | global_average_thermosteric_sea_level_change | 1 |
| Sea water potential temperature | $^{o}$C | sea_water_potential_temperature | 1 |
| Sea surface temperature | $^{o}$C | sea_surface_temperature | 2 |
| Sea water salinity | Psu | sea_water_salinity | 1 |
| Water flux into sea water from iceberg | kg m$^{-2}$ s$^{-1}$ | water_flux_into_sea_water_from_icebergs | 2 |
| Water flux into sea water from ice sheets | kg m$^{-2}$ s$^{-1}$ | water_flux_into_sea_water_from_land_ice | 3 |

Table A3: Data on the LImonant, LImongre, LIyrant or LIyrgre Tables needed to capture the dynamical ice sheet model realm. These fields are saved on the ice sheet grid and contain monthly or yearly output. Tier indicate priority of variable: Mandatory (1), Desirable (2), Experimental (3). These variables are requested for models participating in the *XXX-withism, ism-XXX-self* and *ism-XXX-std* experiments (Table 2). Flux variables are defined positive when the process adds mass or energy to the ice sheet and negative otherwise.

| Long name (netCDF) | Units | Standard name (CF) | Tier |
|---|---|---|---|
| Ice sheet altitude | m | surface_altitude | 1 |
| Ice sheet thickness | m | land_ice_thickness | 1 |
| Bedrock altitude | m | bedrock_altitude | 1 |
| Bedrock geothermal heat flux | $W\ m^{-2}$ | upward_geothermal_heat_flux_at_ground_level _in_land_ice | 3 |
| Land ice calving flux | $kg\ m^{-2}\ s^{-1}$ | land_ice_specific_mass_flux_due_to_calving | 3 |
| Land ice vertical front mass balance flux | $kg\ m^{-2}\ s^{-1}$ | land_ice_specific_mass_flux_due_to_calving_a nd_ice_front_melting | 2 |
| Surface mass balance and its components | $kg\ m^{-2}\ s^{-1}$ | see Table A1 | 1 |
| Basal mass balance of grounded ice sheet | $kg\ m^{-2}\ s^{-1}$ | land_ice_basal_specific_mass_balance_flux | 2 |
| Basal mass balance of floating ice shelf | $kg\ m^{-2}\ s^{-1}$ | land_ice_basal_specific_mass_balance_flux | 2 |
| X-component of land ice surface velocity | $m\ yr^{-1}$ | land_ice_surface_x_velocity | 1 |
| Y-component of land ice surface velocity | $m\ yr^{-1}$ | land_ice_ surface_y_velocity | 1 |
| Z-component of land ice surface velocity | $m\ yr^{-1}$ | land_ice_ surface_upward_velocity | 2 |
| X-component of land ice basal velocity | $m\ yr^{-1}$ | land_ice_basal_x_velocity | 1 |
| Y-component of land ice basal velocity | $m\ yr^{-1}$ | land_ice_basal_y_velocity | 1 |
| Z-component of land ice basal velocity | $m\ yr^{-1}$ | land_ice_basal_upward_velocity | 2 |
| X-component of land ice vertical mean velocity | $m\ yr^{-1}$ | land_ice_vertical_mean_x_velocity | 2 |
| Y-component of land ice vertical mean velocity | $m\ yr^{-1}$ | land_ice_vertical_mean_y_velocity | 2 |

| | | | |
|---|---|---|---|
| Land ice basal drag | Pa | magnitude_of_basal_drag_at_land_ice_base | 3 |
| Surface temperature | K | surface_temperature | 1 |
| Temperature at the top of ice sheet model | K | temperature_at_top_of_ice_sheet_model | 1 |
| Basal temperature of grounded ice sheet | K | temperature_at_base_of_ice_sheet_model | 1 |
| Basal temperature of floating ice shelf | K | temperature_at_base_of_ice_sheet_model | 1 |
| Land ice area fraction | 1 | land_ice_area_fraction | 1 |
| Grounded ice area fraction | 1 | grounded_ice_sheet_area_fraction | 1 |
| Floating ice sheet area fraction | 1 | floating_ice_sheet_area_fraction | 1 |
| Surface snow area fraction | 1 | surface_snow_area_fraction | 2 |

**Scalar outputs / Integrated measures**

| | | | |
|---|---|---|---|
| Ice mass | kg | land_ice_mass | 2 |
| Ice mass not displacing sea water | kg | land_ice_mass_not_displacing_sea_water | 2 |
| Area covered by grounded ice | $m^2$ | grounded_land_ice_area | 3 |
| Area covered by floating ice | $m^2$ | floating_ice_shelf_area | 3 |
| Total SMB flux | $kg\ s^{-1}$ | tendency_of_land_ice_mass_due_to_surface_mass_balance | 3 |
| Total BMB flux | $kg\ s^{-1}$ | tendency_of_land_ice_mass_due_to_basal_mass_balance | 3 |
| Total calving flux | $kg\ s^{-1}$ | tendency_of_land_ice_mass_due_to_calving | 3 |

**Figures and Figure Captions**

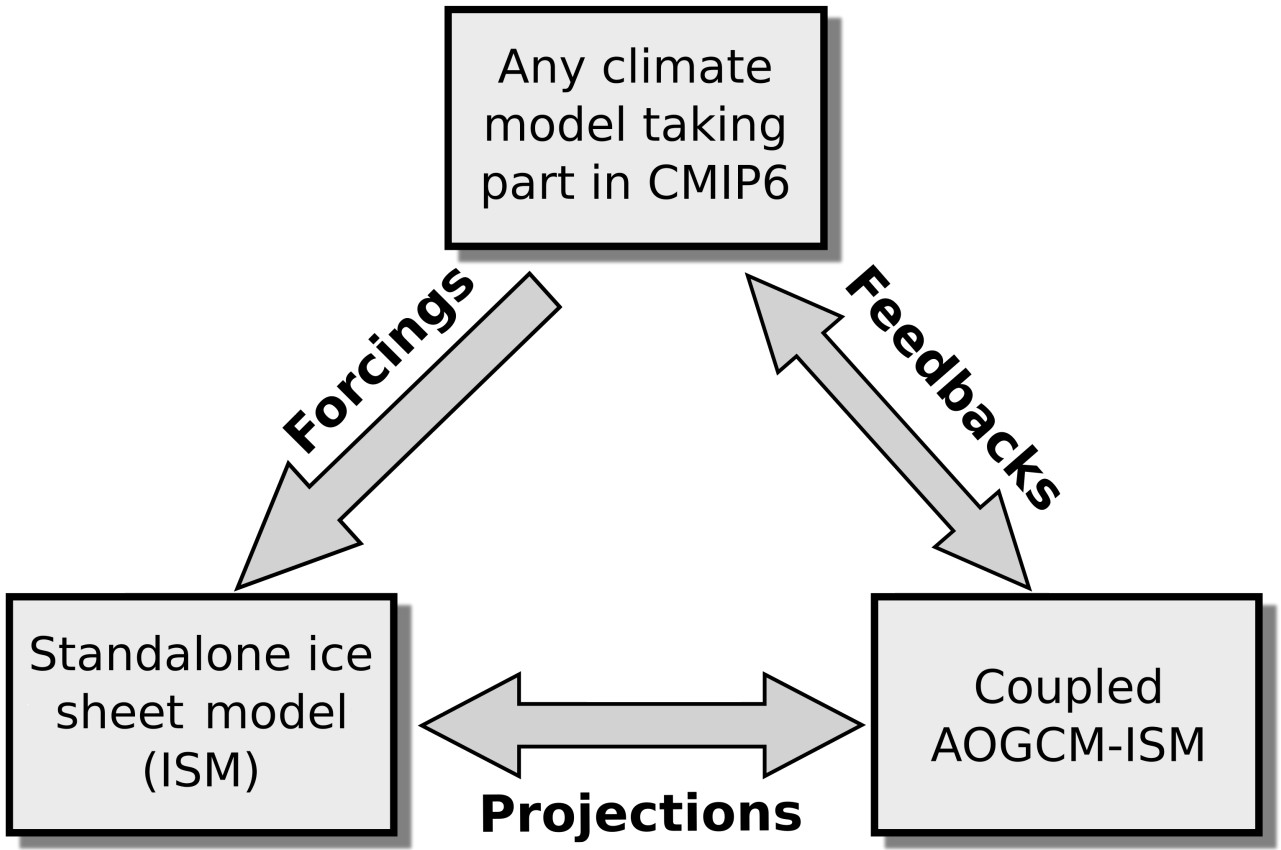

Figure 1: Overview of the ISMIP6 effort designed to obtain forcing from climate models, project sea-level contributions using ice sheet models, and explore ice sheet-climate feedbacks.

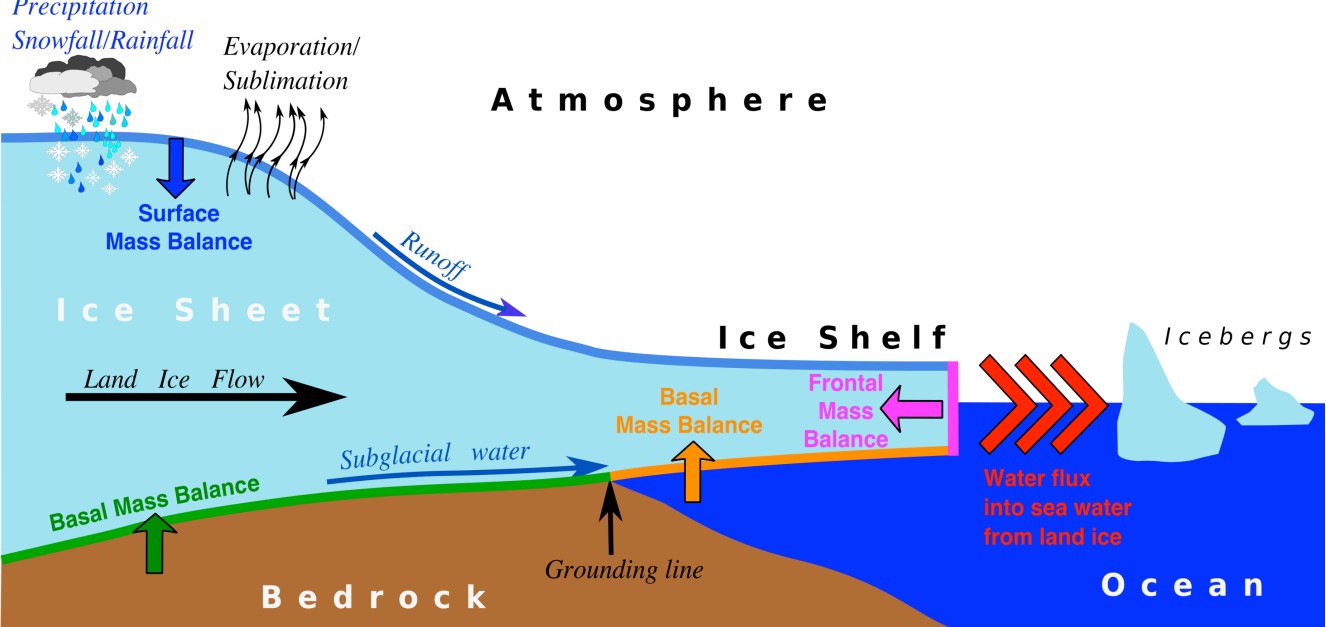

**Figure A1: Illustration of the mass change of ice sheets and key processes that are specific to ice sheet model evaluation or forcing. See text for details.**

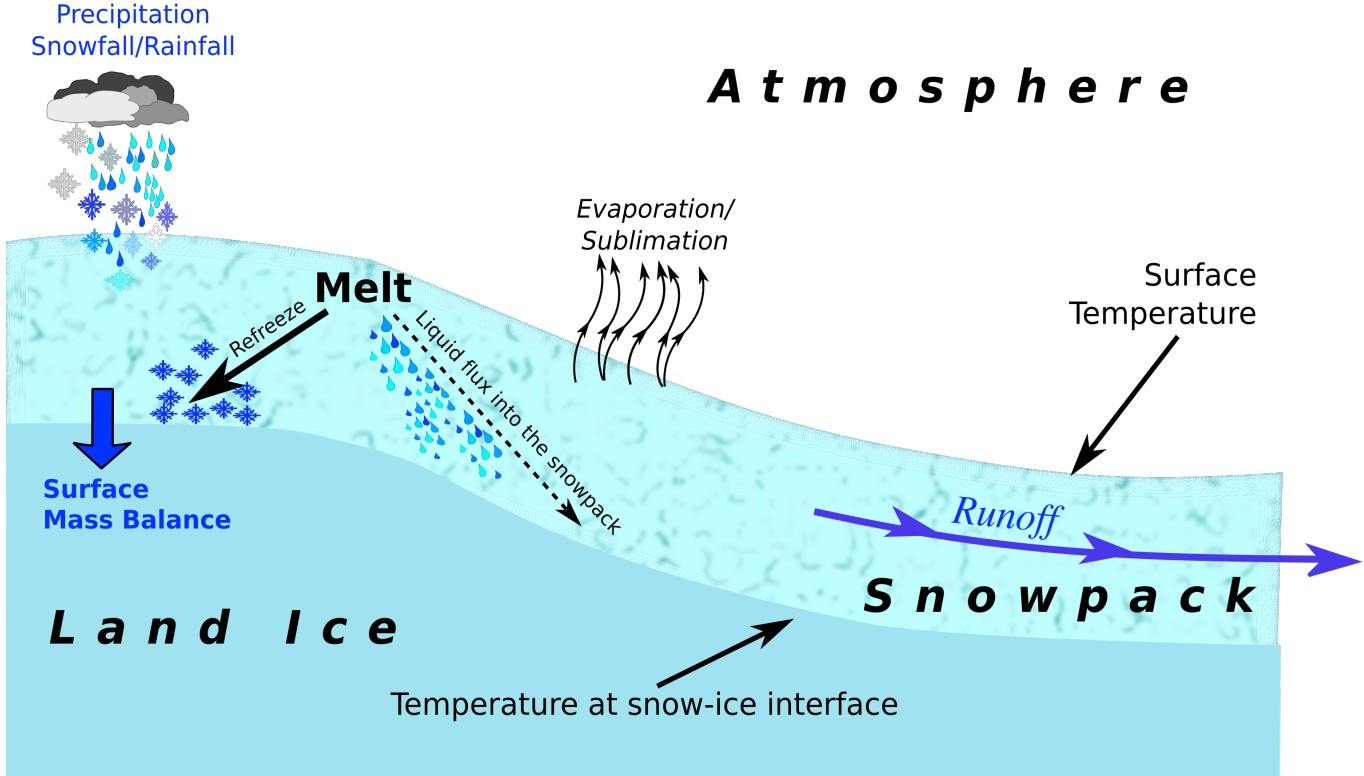

**Figure A2: Illustration of key processes needed to compute atmospheric forcing for ice sheet models, and to evaluate the surface mass balance simulated by climate models.**

