# Peer review of "Ice Sheet Model Intercomparison Project (ISMIP6) contribution to CMIP6"

_Geoscientific Model Development, 2016_

## Short Comment (SC1) · 1 Jul 2016

I enjoyed reading this manuscript clearly describing ISMIP6 initiative. I would however suggest to add a table similar to table 2. I think, it would be appropriate to list the ice sheet models with the corresponding research groups who expressed an interest in participating in this MIP.

---

## Referee Comment (RC1) · C. Rodehacke (Referee) · 13 Jul 2016

Review of Nowicki, Sophie M.J. and others (Geoscientific Model Development, gmd-2016-105)

The manuscript of Nowicki and others describes the foundation and reasoning of the Ice Sheet Model Intercomparison Project (ISMIP6) and its relation the coming CMIP6 exercise.

Under the frame of the Climate and Cryosphere (CliC) and World Climate Research Project (WCRP) model intercomparisons have been and will be a tool to project the future evolution of the earth's climate system. The understanding of how massive land ice masses — such as Antarctica and the Greenland ice sheet — will melt under a changing climate and contribute to an already globally raising sea level is crucial in the

context of climate adaptation and mitigation efforts. The manuscript clearly contrasts the difference between various model experiment setups and motivates the usage of these experiments. It also highlights the relationship to former and coming "traditional" CMIP experiments. The manuscript will certainly act as the reference for the described ISMIP6 exercise.

The manuscript is very well written, has a clear structure and all tables and figures are necessary and well prepared. It was a pleasure to review this manuscript. I hope that the manuscript could be published soon, because I will be extremely helpful to have this information for the involved groups as well as the wider CMIP6 audience.

**I recommend the publication of the manuscript after few minor corrections.**

A detailed list of comments including the text above is given in the attached pdf-file.

Please also note the supplement to this comment:
http://www.geosci-model-dev-discuss.net/gmd-2016-105/gmd-2016-105-RC1-supplement.pdf

**Supplement:**

**Review of Nowicki, Sophie M.J. and others (Geoscientific Model Development, gmd-2016-105)**

The manuscript of Nowicki and others describes the foundation and reasoning of the Ice Sheet Model Inter-comparison Project (ISMIP6) and its relation the coming CMIP6 exercise.

Under the frame of the Climate and Cryosphere (CliC) and World Climate Research Project (WCRP) model intercomparisons have been and will be a tool to project the future evolution of the earth's climate system. The understanding of how massive land ice masses --- such as Antarctica and the Greenland ice sheet --- will melt under a changing climate and contribute to an already globally raising sea level is crucial in the context of climate adaptation and mitigation efforts. The manuscript clearly contrasts the difference between various model experiment setups and motivates the usage of these experiments. It also highlights the relationship to former and coming "traditional" CMIP experiments. The manuscript will certainly act as the reference for the described ISMIP6 exercise.

The manuscript is very well written, has a clear structure and all tables and figures are necessary and well prepared. It was a pleasure to review this manuscript. I hope that the manuscript could be published soon, because I will be extremely helpful to have this information for the involved groups as well as the wider CMIP6 audience.

**I recommend the publication of the manuscript after few minor corrections.**

**Major Issues**

None

**Minor Issues**

First I give general comments and afterwards specific comments.

**General Comments**

In the manuscript I prefer a consistent spelling of either "preindustrial" or "pre-industrial".

Since this manuscript will probably be a reference for the research groups participating, you may add a small table with the essential deadlines to the section 3.4 "Prioritization of experiments and timing". It could act like a "checkbox" table. Please share your thoughts about it.

In the tables A1, A2, and A3 or/and in the appendix A "Variable Request", I would like to see the definition of the flux direction (sign convention). Is ablation (ice loss) a positive or negative flux? Please clarify the text and add (also) please a short remark to the corresponding table captions.

**Specific comments**

In the following specific comments are made, where "P3L23" means line 23 on page 3, for instance.

P4L21: The term ''offline'', may not be known to a general audience. You may rephrase item ii): 'standalone dynamic ice sheet models (ISMs) that are driven by provided forcing fields ("offline").'

P4L26: I personally find the suddenly appearing "XXX" confusing. You may add the sub-clause: where XXX stands for different forcing scenarios as described later.

P5L16: In the bracket the term "fixed is vague. You may mean "reference ice sheet extent and topography"? If so, please specify a possible reference for illustration.

P5L27: You may add "pre-industrial" to obtain:"… is meant to capture the pre-industrial quasi-equilibrium state of the climate system."

P6L25: You state to use the same ice sheet initial condition, which comes from the coupled *XXX-withism* run, for the *XXX-withism* and the *ism-XXX-self* simulations. Since the geometry of the ice sheet could be quite different between the standard AOGCM and the coupled AOGCM-ISM in terms of ice sheet elevation, for instance, the starting conditions and climatic forcing of the standard AOGCM may not be consistent with the *XXX-withism* ice sheet. Hence the forced ice sheet may show a considerable drift and ultimately this drift overprints the actually wanted impact of the difference between coupled vs uncoupled simulations for simulations of about 150 years. Could be please be so kind and comment.

P7L8: I'm unsure if a 'the' is missing:" The choice of the ice sheet model, …. ."

P7L14: You may add:" However, any correction… ."

P7L22-23: I'm skeptical about the implicit statement that internally computed surface mass balance (SMB) calculations are automatically mass and energy conserving while externally computed SMB are not. A wrong regridding from the probably coarse atmospheric grid to the finer ice sheet model grid could break the conservation (Fischer et al., 2014), regardless if the computation is performed inside or outside the AOGCM. I would like to suggest a more general phrasing such as:"… SMB is obtained from energy-based method that conserves mass and energy. It facilitates interpretation of the drivers of SMB variability and change …."

P7L29: I guess I understand what is meant by a "realistic" state, but I would claim that this state is uncertain for the pre-industrial era and that an ice sheet state that is consistent with the driving AOGCM climate is more important. Hence you may agree in replacing "… to produce a realistic non-drifting coupled state" with "… to produce a consistent non-drifting coupled state".

P7L29: As mentioned above, the pre-industrial state is likely different than the contemporary observed state. Hence you may add the following sub-clause:"... to the pre-industrial (1850) climate, which is different from the contemporary state (Kjeldsen et al., 2015)."

P10L29: I would like to suggest a slight clarification:" … temperature changes using the relation of Rignot and Jacobs (2002) of 10 yr$^{-1}$ °C$^{-1}$ for temperatures above the actual ocean's freezing temperature." Please use "°C" instead of "C".

P11L11: For the first time initMIP is mentioned. Please either introduce it or mention where it is described below.

P11L12: I'm sorry, but I do not understand or could not find the referenced section provided in the bracket. Please clarify. In addition this information seems to disagree with the information in the bracket below (P11L22).

P11L23: In my humble opinion a 1% raising atmospheric $CO_2$ concentration has not a linear trend. Hence I suggest:" considers a 1%/year atmospheric $CO_2$ concentration rise until quadrupled concentration and stabilization thereafter."

P11L27: Here you may add:" … to pre-industrial conditions, which is probably weaker, constrained than the contemporary state."

P11L31: Is the leading "to" needed?

P12L28: You may indicate that some groups have provided longer runs by stating:"… each run for at least one hundred years."

P13L8/9: I'm not sure but maybe a pronoun is missing:"… geometric changes in these forward experiments." Please check.

P14L31: I would like to suggest to add a more recent citation for the HIRHAM model: (Langen et al., 2015; Lucas-Picher et al., 2012)

P15L4: For the Greenland ice sheet a very valuable set of observations in the ablation zone comes from the PROMICE network. Therefore I suggest the following change:" … known as the GC-Net (Steffen and Box, 2001), PROMICE network with a focus on the ablation zone (Ahlstrøm et al., 2008)".

P15L28: In addition to the common glaciological estimates I would like to add the following:" … can be compared with glaciological estimates of ice shelf melting around Antarctica (Rignot et al., 2013; Depoorter et al., 2013) as well as independent tracer-oceanographic estimates (Loose et al., 2009; Rodehacke et al., 2006)."

P17L27: You may highlight the coupled simulations in the conclusion by extending:"… no dynamic ice sheets, coupled AOGCM-ISM, and standalone…."

P19L16: Some glaciologists may feel more welcome when instead 'lost' the common term 'ablation' is also used. What do you thing about:" … and ablation to the ocean by either calving or melting."

**Tables**

Here I refer to the table number

Table 2: Please correct the entry for the EC-Earth model. Here the Danish Meteorological Institute (DMI) in Denmark has expressed the interest in the name of the entire consortium.

Table A1, A2, A3: I believe the fractional quantities refer to the total ice covered area. Please clarify and mention it in the table caption.

Table A1, A2, A3: Please indicate in the table caption the sign convention of the fluxes, as already mentioned the general comments section above.

Table A2: Please clarify what is the base line of the "Global Average Thermosteric Sea Level Change"? Is it the beginning of each individual simulation or since the historical period started in 1850, for instance?

**Figures**

The figure numbers are given.

Figure A2: Since runoff leaves the snowpack, I would prefer that the arrow points beyond the snowpack.

**References**

Ahlstrøm, A. P., Gravesen, P., Andersen, S. B., van As, D., Citterio, M., Fausto, R. S., Nielsen, S., Jepsen, H. F., Kristensen, S. S., Christensen, E. L., Stenseng, L., Forsberg, R., Hanson, S., Petersen, D. and Team, P. P.: A new programme for monitoring the mass loss of the Greenland ice sheet, Copenhagen, Denmark. [online] Available from: http://www.geus.dk/DK/publications/geol-survey-dk-gl-bull/15/Documents/nr15_p61-64.pdf, 2008.

Fischer, R., Nowicki, S., Kelley, M. and Schmidt, G. A.: A system of conservative regridding for ice–atmosphere coupling in a General Circulation Model (GCM), Geosci. Model Dev., 7(3), 883–907, doi:10.5194/gmd-7-883-2014, 2014.

Kjeldsen, K. K., Korsgaard, N. J., Bjørk, A. A., Khan, S. A., Box, J. E., Funder, S., Larsen, N. K., Bamber, J. L., Colgan, W., van den Broeke, M., Siggaard-Andersen, M.-L., Nuth, C., Schomacker, A., Andresen, C. S., Willerslev, E. and Kjær, K. H.: Spatial and temporal distribution of mass loss from the Greenland Ice Sheet since AD 1900, Nature, 528(7582), 396–400, doi:10.1038/nature16183, 2015.

Langen, P. L., Mottram, R. H., Christensen, J. H., Boberg, F., Rodehacke, C. B., Stendel, M., van As, D., Ahlstrøm, A. P., Mortensen, J., Rysgaard, S., Petersen, D., Svendsen, K. H., Aðalgeirsdóttir, G. and Cappelen, J.: Quantifying energy and mass fluxes controlling Godthåbsfjord freshwater input in a 5 km simulation (1991-2012), J. Clim., 28(9), 3694–3713, doi:10.1175/JCLI-D-14-00271.1, 2015.

Loose, B., Schlosser, P., Smethie, W. M. and Jacobs, S.: An optimized estimate of glacial melt from the Ross Ice Shelf using noble gases, stable isotopes, and CFC transient tracers, J. Geophys. Res., 114(C8), C08007, doi:10.1029/2008JC005048, 2009.

Lucas-Picher, P., Wulff-Nielsen, M., Christensen, J. H., Aðalgeirsdóttir, G., Mottram, R. and Simonsen, S. B.: Very high resolution regional climate model simulations over Greenland: Identifying added value, J. Geophys. Res., 117(D02108), 16pp, doi:10.1029/2011JD016267, 2012.

Rodehacke, C. B., Hellmer, H. H., Huhn, O. and Beckmann, A.: Ocean/ice shelf interaction in the southern Weddell Sea: results of a regional numerical helium/neon simulation, Ocean Dyn., 57(1), 1–11, doi:10.1007/s10236-006-0073-2, 2006.

---

## Referee Comment (RC2) · X. Asay-Davis (Referee) · 19 Jul 2016

**Review of Nowicki et al., Ice Sheet Model Intercomparison Project (ISMIP6) contribution to CMIP6**

The manuscript provides a valuable summary of the set of experiments—in coupled climate models both with and without ice-sheet components and in standalone ice-sheet models—and compelling motivation for why these experiments will be useful for exploring the role of Greenland and Antarctic Ice Sheets in the climate system, particularly as related to sea-level change.

**General Comments**

The manuscript is well written. Over all, I find the description of the experiments to be quite clear and well thought through. Clearly a commendable effort has gone into designing these experiments. The structure is clear with a few minor exceptions detailed below. The figures provide valuable visual cues to the structure of the experiments as well as the physical processes included in participating models. However, some of the figures are not yet publication quality and could use some additional attention (again, as detailed below).

As my area of expertise is more in ice sheet-ocean coupling and ocean modeling, rather than ice sheet modeling, my most detailed comments relate to ice-ocean interactions. I find that the discussion of potential methods for incorporating melt rates and/or temperature data from the ocean components of AOGCMs as well as the potential used of melt parameterizations needs some further elaboration, as elaborated in the specific comments.

Some of the discussion of how the time ranges of the "XXX-withism" and "ism-XXX-self" and "ism-XXX-std" differ from those of the standard CMIP6 runs they correspond to was not clear to me. I think this issue applies primarily to the historical runs? As I mention below, perhaps this could be clarified better both in the text and by putting the modified ISM ranges into Table 1, rather than having only the standard CMIP6 ranges.

For future Copernicus manuscripts, consider putting the tables and figures inline rather than at the end. This makes the paper much easier to review and is allowed by Copernicus as of January 2016.

**My recommendation is that the manuscript be published with minor corrections.**

**Specific Comments**

p. 3 l. 11: You may wish to define SRES and RCP the first time you refer to them, though these acronyms will be familiar to most readers.

p. 4 l. 26: Like Reviewer 1 (Christian Rodehacke), I felt that the XXX convention should be explicitly defined, even though it is likely obvious to the reader.

p. 5 l. 2: It might be worth mentioning here that you will be discussing the method used to assess and evaluate the AGCM results in Sec. 4 (e.g. "...is to assess and evaluate (using metrics discussed in Sec. 4) CMIP atmosphere..."). During my first reading of the manuscript, I missed that the details of the analysis would come later (though you state it on p. 3 l. 19) and I was expecting at least some sense of what fields, metrics, etc. would be used in this analysis.

p. 8 l. 14: "The Tier 2 experiments..." You haven't yet introduced the tiers for the different experiments at this point in the text. If you can avoid referring to Tier 2 here by giving those experiments some other descriptor, that would save the awkwardness of needing to introduce the tiers here, rather than later where they seem to fit best.

p. 8 l. 30-32: It is not entirely clear what "it" refers to in this sentence, presumably "accurate treatment of ice-ocean interactions"? More importantly, it seems to me that there is little doubt that accurate treatment of ice-ocean interactions requires moving boundaries in the ocean model. Just as parameterizing, rather than explicitly simulation, the circulation in ice-shelf cavities and resulting melt rates leads to inaccuracies, there can be little doubt that ignoring changes in cavity geometry (or parameterizing changes in melt rates) as the ice sheet evolves will lead to inaccuracies. All that is to say that "may" should be replaced with something stronger like "will likely".

p. 9 l. 21-23: I would suggest moving "based on an initial analysis of AOGCM simulation[s] of ice-sheet climate" to the beginning of the sentence for clarity. That way, it is hopefully clear that you are identifying the experiments based on the initial analysis, rather than that the ISMs are performing experiments based on the initial analysis. Also, maybe again here you could say that the criteria for determining which AOGCM results are "best" (i.e. chosen for the small subset of experiments) will be discussed in Sec. 4.

p. 10 l. 1-2: "...mismatch in spatial resolution over which SMB varies and that is used by AOGCMs". This phrase is confusing to me. Perhaps "..mismatch between the spatial resolution of AOGCMs and the characteristic length scale of variations in SMB"?

p. 10. l. 8: The use of RCMs as intermediaries between AOGCMs and ice-sheet models also adds ambiguity about which biases are introduced by the AOGCMs and which by the RCMs, does it not?

Paragraph starting at p. 10 l. 26: Presumably, an effective melt parameterization would need to account for both the phenomena you outline in this paragraph (and probably more). It would need to make use of ocean temperature (and probably salinity) as a function of depth somewhere near the calving front each ice shelf and also the depth of the ice draft within the cavity. More sophistication would be nice (e.g. accounting for faster ocean flow with steeper ice-draft slope) but is still a topic of ongoing research.

p. 10 l. 27-28: I do not think that Rignot and Jacobs used surface temperature for their relationship, but rather ocean-bottom temperature close to the calving front. (However, they do not state their method of obtaining the temperature explicitly in their paper, at least as far as I could tell.) Also, this relation is only calibrated for melt rates at the GLs and likely is missing important nonlinearities (Holland et al. 2008).

p. 10 l. 29-30: "...that depends on the ocean temperate at the closest grid cell..." At what depth would the temperature be taken? Hopefully at or near the ocean bottom. Or better yet as a profile of depth.

p. 10 l. 30-31: "If none of the CMIP6 ocean models are suitable" Can you be more specific about how "suitable" is defined (or refer to Sec. 4 and make sure you define there how you determine whether ocean results are suitable)?

p. 10 l. 31-32: "prescribe a melt parameterization that depends simply on the ice shelf draft". I (and other ocean modelers) feel that this is a poor choice (perhaps very much so) for a couple of reasons: 1)

The thermal forcing (or thermal driving – the difference between the freezing point and the "ambient" ocean temperature, however "ambient" is defined) plays at least as important a role as the depth of the ice draft, so that differences between "warm" and "cold" ice shelves cannot be ignored.  2) Such parameterizations have only been used in small regions, where their coefficients have been calibrated to local thermal conditions, not over the whole of Antarctica.

p. 11 l. 1: "oceanic anomalies (basal mass balance and basal temperatures)" I do not see how these can be generated, independent of ice basal topography (ice draft) if a parameterization is being used. Instead, perhaps coefficients in the parameterization as functions of time could be provided, from which melt rates could be computed given an ice draft.

Paragraph starting at p. 11 l. 20: I found this whole paragraph to be very confusing.  Perhaps part of it is that initMIP has not yet been described.  Maybe you could consider reordering the paper so initMIP has been described already at this point?

p. 11 l. 21-22: It is not at all clear to me what these two sentences refer to.  Is the 1990s to 2014 forcing repeated or is just 2014 repeated? Or something else?

p. 11 l. 27: Please elaborate on the challenges of initializing ice sheet models to per-industrial conditions and how this presents challenges that do not allow for the typical historical run.  This is likely not obvious to all of your readers.

p. 11 l 27-29: What does this mean?  What period of time is covered?  Please consider updating Table 1 so the range of times for the various ism simulations is given separately where they differ from the standard CMIP6 simulations.  This would help to clarify the confusing differences in time ranges described in this paragraph compared with those of standard CMIP6.

p. 13 l. 3-5: How will the *abmb* anomaly field be constructed?  How will it be made to conform to differences in grounding lines and calving fronts between different models?  Similarly, how will the SMB anomaly be made to conform to differences in ice sheet extent between models in *asmb*?

p. 13 l. 19-20: Why would it be ideal for the ism experiments to follow the AOGCM experiments with a six-month lag?  Do you perhaps mean "no more than a six-month lag"?  I would think ideally the ISM experiments would follow the AOGCM ones without any lag at all, but a realistic (or perhaps somewhat optimistic) time table would be for a six-month lag.

p. 15 l. 29-30: "regional ocean models (e.g. Timmermann et al. 2012)" FESOM, the model that was the primary focus of this Timmermann paper, is actually a global model with high resolution focused in Antarctica.  Perhaps "regionally focused ocean models" would be more correct?

p. 33 Table 1: Please consider putting the actual start and end year for each ISM simulation that used a range different from the default CMIP6 (as requested previously)?

p. 41 Figures A1: I feel this figure need some cleanup before they look professional enough for publication.  The curves are not lined up very well (black is peaking out from under green).  The blue arrows on the lighter blue surface and green base are not very visible.  The giant gray error for freshwater flux should be given adequate room so it doesn't overlap the ice berg.  Black lines should be anti-aliased and boundaries of the figure should not be jagged (slanted with respect to the figure caption).

p. 42 Figures A2: Blue text (both light and dark) is hard to read on blue background.  The phrase "Liquid flux into the snowpack" should ideally either be entirely within or entirely outside the blue region.

**Typographical Corrections**

p. 5 l. 22: "amip" needs to be punctuated differently.  Perhaps "The Atmospheric Model Intercomparison Project (*amip*; Gates et al. 1999) simulation allows..."

p. 6. l. 18: "four ISMs simulations" → "four ISM simulations"?

p. 9 l. 22: "AOGCM simulation" → "AOGCM simulations"

p. 9 l. 28: "...cannot be made however we list..." → "...cannot be made. However, we list..."

p 10 l. 5: RCMS → RCMs.  Also, a verb is missing in "to SMB", perhaps, "to simulate SMB"?

p. 10 l. 18: "the SMB lapse rate obtained" I would remove the word "obtained".  It is not needed.

p. 11 l. 8: "the dynamic response output" I would remove the word "output".

p. 11 l. 11: "Ice Sheet-Ocean-Model" → "Ice Sheet-Ocean Model"

p. 11 l. 12: "Sect. 3.3). )" there is some extra punctuation here.  I think just the one end parenthesis is needed.  Also, should this be Sect. 3.3.2?

p. 11 l. 23-24: I would change "our" to "the" at the beginning of both these sentences.

p. 11 l. 32: "ice sheets evolution" → "ice sheet evolution"

p. 12 l. 1: "at least by 4 meters" → "by at least 4 meters"

p. 12 l. 2: "from the ism-lig-std" → "from ism-lig127k-std"

p. 12 l. 30: "Antarctic Ice Sheets" → "Antarctic Ice Sheet"

p. 15 l. 28: "As regional" → "Just as regional"

p. 19 l. 25-26: "not explicitly asked to minimize the data request" → "not specifically requested as an output variable in order to reduce the size of the data files" (or something similar – the original phrasing is not very clear)

p. 20 l. 26-39: Note that Asay-Davis et al. has been accepted in GMD so please don't forget to update the reference when the time comes.

p. 31 l. 5 and 18: In 2 references, Vizcaíno is spelled without the accent mark while in three it is with the accent mark.  This may be an issue with the respective journals but it looks strange when these articles are cited close together.

**Reference**

Holland, P. R., Jenkins, A., & Holland, D. M. (2008). The Response of Ice Shelf Basal Melting to Variations in Ocean Temperature. Journal of Climate, 21(11), 2558–2572. http://doi.org/10.1175/2007JCLI1909.1

---

## Short Comment (SC2) · 21 Jul 2016

**Comments from CMIP Panel**

The CMIP Panel is undertaking a review of the CMIP6 GMD special issue papers to ensure a level of consistency among the invited contributions, also in answering the key questions that were outlined in our request to submit a paper to all co-chairs of CMIP6-Endorsed MIPs. We very much welcome the important contribution from ISMIP6 to the CMIP6 special issue, below are a few comments:

Please consistently use the term *'CMIP6-Endorsed MIPs'* when you refer to other MIPs that are endorsed by CMIP6.

Please ensure consistency of the experiment names and abbreviations with the CMIP6

overview paper (Eyring et al., 2016).

Please ensure that all ISMIP6 'experiment_ids' and 'sub_experiment_ids' are consistent with those used in the CMIP6 data request and experiment table, and with the CMIP6 terminology (see email exchange with Karl Taylor).

Please ensure that the 'source_id' for the offline models is compliant with CMIP6 terminology.

p.5,l1ff. Section 3.1

- Is there a more intuitive title that actually describes what is envisaged here? *Use* seems vague.

- The last two paragraphs (l21ff and p6,l1ff) on the definition and explanation of the DECK, CMIP6 historical simulation, ScenarioMIP, and PMIP experiments should be deleted since they are already defined elsewhere in this special issue. It seems sufficient to simply refer to the other papers.

- Would it be possible to list some observations that could be used to assess the models (l15ff)?

- Possibly merge the first paragraph with Section 4.1 which is on the same subject and seems repetitive as it stands now with bits of information scattered across these two sections.

p7,l19ff: please add to the text that both the downscaling method and the spin-up should be well documented. Also it might be good to expand a bit on the methods used for the spinup.

p8,l8: '*then holds concentrations fixed for an additional two to four centuries.*'. This is not how the *1ptCO2* experiment is defined. Note that in contrast to previous definitions, the experiment has been simplified so that the 1% CO2 increase per year is applied

throughout the entire simulation rather than keeping it constant after 140 years as in CMIP5, see Section A1.4 in Eyring et al. (2016).

p8,l21: Here you encourage the extension of the projections until 2300 which is certainly a valid addition when it comes to the assessment of future sea level change. It is however only recommended, whereas in Table 1 this experiment is listed until 2300. Please could you make this consistent, e.g. either make it mandatory or - similar to ScenarioMIP (e.g. SSP5-8.5-Ext) - separate the two simulations in Table 1 into one that goes until 2100 and one that extends to 2300, and additionally list the Tier?

p9,l5ff: Section 3.3: Could you confirm that all the output from the offline and standalone ISM experiments is conform to the output requirements of CMIP6? If ISMIP6 relaxes this requirement on output for some of its offline experiments, then those experiments should be considered not a part of CMIP6 (and therefore not listed in this paper). They could be described elsewhere. Our experience with past MIPs has been that initially the threshold effort required for standardizing data output (CMORization) is perceived as an obstacle by many groups, but time and experience has shown that this effort is well worth it. We have found that only standardized data gets widely used by the community, and the analysis of that data, especially by researchers outside the major modeling centers, has been central to CMIP's success.

p9,l13ff: '*A key concern is that ISMIP6 assess uncertainty associated with both emission scenario and the AOGCMs' simulation of these scenarios. To this end, we anticipate identifying a subset of the CMIP6 AOGCM ensemble for use as ISM forcing which captures the full range of potential ice-sheet forcing.*' This paragraph is too vague and the sentence seems contradicting - first a subset is selected and this should then be the full range? Please clarify. Please also add more explanation on how this selection process is done and why it is necessary.

l18,p15: Data availability:

- Please delete '*the majority*' in the first sentence. All CMIP6 simulations will be distributed through the ESGF and non-CMIP6 experiments shouldn't be described in this paper.

- Could you please add the following additional sentence after the first sentence? '*In order to document CMIP6's scientific impact and enable ongoing support of CMIP, users are obligated to acknowledge CMIP6, the participating modelling groups, and the ESGF centres (see details on the CMIP Panel website at http://www.wcrp-climate.org/index.php/wgcm-cmip/about-cmip).*'

Table 1: The table lists experiments that are defined by ISMIP6 and experiments that are already defined elsewhere in CMIP6, this is confusing.

- Suggest to remove all experiments that are already defined elsewhere from this table (i.e., please remove the entire row for *amip*, *abrupt-4xCO2*, *historical*, *ssp5-8.5*, and *lig127k*).

- The titles of each category could be more specific by for example saying 'ISMIP6 DECK experiments' or something similar.

- Please could you also add a column that shows the Tier for each experiment?

- There could be another category that lists the ISMIP6 offline experiments from Section 3.3 if they are proposed to be part of CMIP6 (in which case the output has to be compliant with CMIP6 standards, see above).

Table 2:

- is it possible to add a column with some specifics on the ice sheet models used and a reference if available?

- GFDL and MPI-ESM were two more models that initially indicated interest in participating but are not listed?

- Except for CanESM that only participates in the diagnostic part of ISMIP6, are all other models listed using fully coupled ice sheet models, or are some of the models listed only contributing with standalone ice sheet models? Maybe this is not fully decided yet?

- Maybe it would be good to also list in a similar manner the standalone ice sheet models?

Tables A1-A3: This is a very helpful overview of the variables requested by ISMIP6 but it would be good to clarify either in the caption or in a separate column from which CMIP6 experiments these variables are requested.

Table A2: Some additional information from the models is required to regrid the ocean data to standard grids. OMIP is proposing a weights file that model groups should provide to enable regridding from the native grid to one or two CMIP6 standard grids. Please refer to Griffies et al. (2016) and follow the same procedure for the ISMIP6 Omon requests if regridding is required.

Reference:

Eyring, V., Bony, S., Meehl, G. A., Senior, C. A., Stevens, B., Stouffer, R. J., and Taylor, K. E.: Overview of the Coupled Model Intercomparison Project Phase 6 (CMIP6) experimental design and organization, Geosci. Model Dev., 9, 1937-1958, doi:10.5194/gmd-9-1937-2016, 2016.

Griffies, S. M., Danabasoglu, G., Durack, P. J., Adcroft, A. J., Balaji, V., Böning, C. W., Chassignet, E. P., Curchitser, E., Deshayes, J., Drange, H., Fox-Kemper, B., Gleckler, P. J., Gregory, J. M., Haak, H., Hallberg, R. W., Hewitt, H. T., Holland, D. M., Ilyina, T., Jungclaus, J. H., Komuro, Y., Krasting, J. P., Large, W. G., Marsland, S. J., Masina, S., McDougall, T. J., Nurser, A. J. G., Orr, J. C., Pirani, A., Qiao, F., Stouffer, R. J., Taylor, K. E., Treguier, A. M., Tsujino, H., Uotila, P., Valdivieso, M., Winton, M., and Yeager, S. G.: Experimental and diagnostic protocol for the physical component of the

CMIP6 Ocean Model Intercomparison Project (OMIP), Geosci. Model Dev. Discuss., doi:10.5194/gmd-2016-77, in review, 2016.

With many thanks for your ongoing efforts in the CMIP6 process.

The CMIP Panel

---

## Author Comment (AC2) · 2 Sep 2016

**Author's response to comments from C Rodehacke (Referee 1)**

**The manuscript of Nowicki and others describes the foundation and reasoning of the Ice Sheet Model Intercomparison Project (ISMIP6) and its relation the coming CMIP6 exercise.**

**Under the frame of the Climate and Cryosphere (CliC) and World Climate Research Project (WCRP) model intercomparisons have been and will be a tool to project the future evolution of the earth's climate system. The understanding of how massive land ice masses — such as Antarctica and the Greenland ice sheet — will melt under a changing climate and contribute to an already globally raising sea level is crucial in the context of climate adaptation and mitigation efforts. The manuscript clearly contrasts the difference between various model experiment setups and motivates the usage of these experiments. It also highlights the relationship to former and coming "traditional" CMIP experiments. The manuscript will certainly act as the reference for the described ISMIP6 exercise.**

**The manuscript is very well written, has a clear structure and all tables and figures are necessary and well prepared. It was a pleasure to review this manuscript. I hope that the manuscript could be published soon, because I will be extremely helpful to have this information for the involved groups as well as the wider CMIP6 audience.**

**I recommend the publication of the manuscript after few minor corrections.**

**A detailed list of comments including the text above is given in the attached pdf-file**.

We thank the reviewer for the helpful comments. We have now revised our manuscript in light of these and other comments that we have received. A point-by-point reply is given below.

**Major Issues**
**None**

**Minor Issues**
**First I give general comments and afterwards specific comments.**

**General Comments**
**In the manuscript I prefer a consistent spelling of either "preindustrial" or "pre-industrial".**

We have checked the manuscript and used a consistent spelling of "pre-industrial"

**Since this manuscript will probably be a reference for the research groups participating, you may add a small table with the essential deadlines to the section 3.4 "Prioritization of experiments and timing". It could act like a "checkbox" table. Please share your thoughts about it.**

We thank the reviewer for this idea. However, the dependence of most of the proposed ISMIP6 experiments on model output from climate modeling centers makes it impossible for us to foresee exact timing of the initiatives at this stage.

**In the tables A1, A2, and A3 or/and in the appendix A "Variable Request", I would like to see the definition of the flux direction (sign convention). Is ablation (ice loss) a positive or**

**negative flux? Please clarify the text and add (also) please a short remark to the corresponding table captions.**

We thank the reviewer for the suggestion. We have added general statements to the table captions to clarify the flux directions.
Table A1: Flux variables are defined positive when the process adds mass or energy to the ice sheet and negative otherwise.
Table A2: Flux variables are defined positive when the process adds mass to the ocean and negative otherwise.
Table A3: Flux variables are defined positive when the process adds mass or energy to the ice sheet and negative otherwise.

**Specific comments**
**In the following specific comments are made, where "P3L23" means line 23 on page 3, for instance.**

**P4L21: The term ''offline'', may not be known to a general audience. You may rephrase item ii): 'standalone dynamic ice sheet models (ISMs) that are driven by provided forcing fields ("offline").'**
We have done the suggested rephrasing.

**P4L26: I personally find the suddenly appearing "XXX" confusing. You may add the sub-clause: where XXX stands for different forcing scenarios as described later.**
We have added the suggested sub-clause.

**P5L16: In the bracket the term "fixed is vague. You may mean "reference ice sheet extent and topography"? If so, please specify a possible reference for illustration.**
We believe 'fixed' is the right term here, referring to the prescribed topography and albedo in classic GCM simulations. Referring to a 'reference' instead would add confusion, since there is really only one state of the ice sheets in these simulations.

**P5L27: You may add "pre-industrial" to obtain:"… is meant to capture the pre-industrial quasi-equilibrium state of the climate system."**
We have done the suggested rephrasing.

**P6L25: You state to use the same ice sheet initial condition, which comes from the coupled XXX-withism run, for the XXX-withism and the ism-XXX-self simulations. Since the geometry of the ice sheet could be quite different between the standard AOGCM and the coupled AOGCM-ISM in terms of ice sheet elevation, for instance, the starting conditions and climatic forcing of the standard AOGCM may not be consistent with the XXX-withism ice sheet. Hence the forced ice sheet may show a considerable drift and ultimately this drift overprints the actually wanted impact of the difference between coupled vs uncoupled simulations for simulations of about 150 years. Could be please be so kind and comment.**
We agree that there is a danger of a drift dominated by any signal forced by the difference between coupled and uncoupled simulations. We could lessen this drift by using SMB anomalies. However, the potential for a drift is one of the reasons why we are not relying on the coupled modeling for our projections, as we expect the results of the standalone ice sheet modeling to be more robust. The coupled modeling is primarily done so that issues (such as this) created by coupling climate and ice sheet models are exposed, and the community can start to work towards resolving them.

The ism-piControl-self will be used to quantify the drift, which we can subtract from the ism-1pctCO2-self, to get the effect of climate change. However, if the drift is large, this may not be satisfactory. We hope that the AOGCM SMB will be realistic enough and that the spin-up geometry between the standard AOGCM and the coupled AOGCM-ISM is not hugely different, and therefore the drift minimal.

We have restructured the manuscript so that the discussion of the spin-up is now before the coupled experiments. In the new structure, the original first three paragraphs have been left unchanged. The fourth paragraph now discusses the spin-up, and it mainly based on the original sixth paragraph, with any change due to reading flow or needed to address comments about spin-up. The fifth paragraph states the ideal of using actual SMB forcing from the AOGCM, and is the bulk of the old fourth paragraph adapted to address your concerns about initial conditions and drift. The remainder of the section is then unchanged.

The fifth paragraph reads "Ideally, the ice sheet model should be forced with the actual SMB computed by the climate model, rather than an SMB corrected to match observed climatology. We accept that there may be biases in the atmospheric or land models that can lead to an unrealistic SMB, which could result in a steady-state ice sheet geometry that differs substantially from present-day observations. However, correcting for these biases can distort the feedbacks between ice sheets and climate that we seek to investigate. We hope to learn from and ultimately reduce these biases, in the same way that biases elsewhere in the simulated coupled climate system are reduced by greater understanding and improved model design. On the other hand, if the geometry of the spun-up ice sheet is greatly different from observations, then the initial ice sheet may be far from steady state with the SMB forcing from the standard, uncoupled AOGCM. As a result, the *ism-piControl-self* experiment could have a large drift that obscures the climate signal. If this is the case, or in general if the spun-up ice sheet in the coupled system is deemed to be too unrealistic, an alternative spin-up method would be to apply SMB anomalies from the AOGCM, superposed on a climatology that yields more realistic equilibrium ice sheet geometry."

**P7L8: I'm unsure if a 'the' is missing:" The choice of the ice sheet model, …. ."**
This has been changed as suggested.

**P7L14: You may add:" However, any correction… ."**
This has been changed as suggested.

**P7L22-23: I'm skeptical about the implicit statement that internally computed surface mass balance (SMB) calculations are automatically mass and energy conserving while externally computed SMB are not. A wrong regridding from the probably coarse atmospheric grid to the finer ice sheet model grid could break the conservation (Fischer et al., 2014), regardless if the computation is performed inside or outside the AOGCM. I would like to suggest a more general phrasing such as:"… SMB is obtained from energy based method that conserves mass and energy. It facilitates interpretation of the drivers of SMB variability and change …."**
We have done the suggested rephrasing.

**P7L29: I guess I understand what is meant by a "realistic" state, but I would claim that this state is uncertain for the pre-industrial era and that an ice sheet state that is consistent with the driving AOGCM climate is more important. Hence you may agree in replacing "… to produce a realistic non-drifting coupled state" with "… to produce a consistent non-drifting coupled state".**
We have done the suggested rephrasing.

**P7L29: As mentioned above, the pre-industrial state is likely different than the contemporary observed state. Hence you may add the following sub-clause:"... to the pre-industrial (1850) climate, which is different from the contemporary state (Kjeldsen et al., 2015)."**
This has been added as suggested.

**P10L29: I would like to suggest a slight clarification:" … temperature changes using the relation of Rignot and Jacobs (2002) of 10 yr-1 °C-1 for temperatures above the actual ocean's freezing temperature." Please use "°C" instead of "C".**
This has been fixed as suggested.

**P11L11: For the first time initMIP is mentioned. Please either introduce it or mention where it is described below.**
Following a suggestion of Reviewer 2, we have moved the initMIP description earlier in the text, which solves this problem.

**P11L12: I'm sorry, but I do not understand or could not find the referenced section provided in the bracket. Please clarify. In addition this information seems to disagree with the information in the bracket below (P11L22).**
Thank you for spotting this, there was a mistake in the referencing. This has been solved along with the changes described in response to the comment before.

**P11L23: In my humble opinion a 1% raising atmospheric CO2 concentration has not a linear trend. Hence I suggest:" considers a 1%/year atmospheric CO2 concentration rise until quadrupled concentration and stabilization thereafter."**
We have done the suggested rephrasing.

**P11L27: Here you may add:" … to pre-industrial conditions, which is probably weaker, constrained than the contemporary state."**
This has been added as suggested.

**P11L31: Is the leading "to" needed?**
This has been kept, as the sentence is "are likely to differ", if the "are" had not been used, we would have indeed removed the "to".

**P12L28: You may indicate that some groups have provided longer runs by stating:"… each run for at least one hundred years."**
This has been added as suggested.

**P13L8/9: I'm not sure but maybe a pronoun is missing:"… geometric changes in these forward experiments." Please check.**
Indeed a pronoun was missing and was added as suggested.

**P14L31: I would like to suggest to add a more recent citation for the HIRHAM model: (Langen et al., 2015; Lucas-Picher et al., 2012)**
This has been added as suggested.

**P15L4: For the Greenland ice sheet a very valuable set of observations in the ablation zone comes from the PROMICE network. Therefore I suggest the following change:" … known as the GC-Net (Steffen and Box, 2001), PROMICE network with a focus on the ablation**

zone (Ahlstrøm et al., 2008)".
This has been added as suggested.

**P15L28: In addition to the common glaciological estimates I would like to add the following:" … can be compared with glaciological estimates of ice shelf melting around Antarctica (Rignot et al., 2013; Depoorter et al., 2013) as well as independent tracer-oceanographic estimates (Loose et al., 2009; Rodehacke et al., 2006)."**
This has been added as suggested.

**P17L27: You may highlight the coupled simulations in the conclusion by extending:"… no dynamic ice sheets, coupled AOGCM-ISM, and standalone…."**
This has been added as suggested.

**P19L16: Some glaciologists may feel more welcome when instead 'lost' the common term 'ablation' is also used. What do you thing about:" … and ablation to the ocean by either calving or melting."**
This has been changed as suggested.

**Tables**
**Here I refer to the table number**
**Table 2: Please correct the entry for the EC-Earth model. Here the Danish Meteorological Institute (DMI) in Denmark has expressed the interest in the name of the entire consortium.**
This has been fixed as suggested.

**Table A1, A2, A3: I believe the fractional quantities refer to the total ice covered area. Please clarify and mention it in the table caption.**
For a gridded data set, these variables conventionally give the fractional area covered by the quantity in a grid cell. No further clarifications needed.

**Table A1, A2, A3: Please indicate in the table caption the sign convention of the fluxes, as already mentioned the general comments section above.**
Additional clarifications have been added to the captions. See also reply to general comment above.

**Table A2: Please clarify what is the base line of the "Global Average Thermosteric Sea Level Change"? Is it the beginning of each individual simulation or since the historical period started in 1850, for instance?**
The data request tables are thought to be universal and would apply equally to e.g. paleo simulations. The standard sea level reference is therefore the beginning of the individual simulation, but may have to be specified for certain cases.

**Figures**
**The figure numbers are given.**
**Figure A2: Since runoff leaves the snowpack, I would prefer that the arrow points beyond the snowpack.**
This has been changed as suggested.

References
Ahlstrøm, A. P., Gravesen, P., Andersen, S. B., van As, D., Citterio, M., Fausto, R. S., Nielsen, S., Jepsen, H. F., Kristensen, S. S., Christensen, E. L., Stenseng, L., Forsberg, R.,

Hanson, S., Petersen, D. and Team, P. P.: A new programme for monitoring the mass loss of the Greenland ice sheet, Copenhagen, Denmark. [online] Available from: http://www.geus.dk/DK/publications/geol-survey-dk-gl-bull/15/Documents/nr15_p61-64.pdf, 2008.

Fischer, R., Nowicki, S., Kelley, M. and Schmidt, G. A.: A system of conservative regridding for ice–atmosphere coupling in a General Circulation Model (GCM), Geosci. Model Dev., 7(3), 883–907, doi:10.5194/gmd-7-883-2014, 2014.

Kjeldsen, K. K., Korsgaard, N. J., Bjørk, A. A., Khan, S. A., Box, J. E., Funder, S., Larsen, N. K., Bamber, J. L., Colgan, W., van den Broeke, M., Siggaard-Andersen, M.-L., Nuth, C., Schomacker, A., Andresen, C. S., Willerslev, E. and Kjær, K. H.: Spatial and temporal distribution of mass loss from the Greenland Ice Sheet since AD 1900, Nature, 528(7582), 396–400, doi:10.1038/nature16183, 2015.

Langen, P. L., Mottram, R. H., Christensen, J. H., Boberg, F., Rodehacke, C. B., Stendel, M., van As, D., Ahlstrøm, A. P., Mortensen, J., Rysgaard, S., Petersen, D., Svendsen, K. H., A›algeirsdóttir, G. and Cappelen, J.: Quantifying energy and mass fluxes controlling Godthåbsfjord freshwater input in a 5 km simulation (1991-2012), J. Clim., 28(9), 3694–3713, doi:10.1175/JCLI-D-14-00271.1, 2015.

Loose, B., Schlosser, P., Smethie, W. M. and Jacobs, S.: An optimized estimate of glacial melt from the Ross Ice Shelf using noble gases, stable isotopes, and CFC transient tracers, J. Geophys. Res., 114(C8), C08007, doi:10.1029/2008JC005048, 2009.

Lucas-Picher, P., Wulff-Nielsen, M., Christensen, J. H., A›algeirsdóttir, G., Mottram, R. and Simonsen, S. B.: Very high resolution regional climate model simulations over Greenland: Identifying added value, J. Geophys. Res., 117(D02108), 16pp, doi:10.1029/2011JD016267, 2012.

Rodehacke, C. B., Hellmer, H. H., Huhn, O. and Beckmann, A.: Ocean/ice shelf interaction in the southern Weddell Sea: results of a regional numerical helium/neon simulation, Ocean Dyn., 57(1), 1–11, doi:10.1007/s10236-006-0073-2, 2006.

We thank the reviewer for the detailed comments. Most of the references have also been included in the manuscript.

---

## Author Comment (AC3) · 2 Sep 2016

**Author's response to comments from X. Asay-Davis (Referee 2)**

**The manuscript provides a valuable summary of the set of experiments—in coupled climate models both with and without ice-sheet components and in standalone ice-sheet models— and compelling motivation for why these experiments will be useful for exploring the role of Greenland and Antarctic Ice Sheets in the climate system, particularly as related to sea-level change.**

We thank the reviewer for the helpful comments. We have now revised our manuscript in light of these and other comments that we have received. A point-by-point reply is given below.

**General Comments**
**The manuscript is well written. Over all, I find the description of the experiments to be quite clear and well thought through. Clearly a commendable effort has gone into designing these experiments. The structure is clear with a few minor exceptions detailed below. The figures provide valuable visual cues to the structure of the experiments as well as the physical processes included in participating models. However, some of the figures are not yet publication quality and could use some additional attention (again, as detailed below).**

Thank you for the detailed comments on the figure. They have been cleaned up accordingly.

**As my area of expertise is more in ice sheet-ocean coupling and ocean modeling, rather than ice sheet modeling, my most detailed comments relate to ice-ocean interactions. I find that the discussion of potential methods for incorporating melt rates and/or temperature data from the ocean components of AOGCMs as well as the potential used of melt parameterizations needs some further elaboration, as elaborated in the specific comments.**

We have altered the manuscript to address your specific comments. However, we are currently not able to be more specific on the way in which the standard ocean forcing will be implemented, as the final choice will depend on the evaluation of CMIP6 ocean runs that have not yet been made by the climate centers. Our goal was therefore to provide a variety of options. This said, ISMIP6 is committed to obtain the best possible forcing for ice sheet models with the help of atmosphere and ocean experts, and in the coming two years, we are organizing a series of workshop to address this issue. The first of these workshops will be held in San Francisco in December 2016, just before AGU.

**Some of the discussion of how the time ranges of the "XXX-withism" and "ism-XXX-self" and "ism-XXX-std" differ from those of the standard CMIP6 runs they correspond to was not clear to me. I think this issue applies primarily to the historical runs? As I mention below, perhaps this could be clarified better both in the text and by putting the modified ISM ranges into Table 1, rather than having only the standard CMIP6 ranges.**

We have altered the manuscript to clarify why it is challenging for standalone ice sheet models to have the same time range as the standard CMIP6 runs, and by including the modified ISM time range in the Table. Indeed the issue primarily applies to the historical run, which in standard CMIP6 runs start from the pre-industrial spin-up or control run. The reason is that pre-industrial spin-up is challenging for ice sheet models, so ice sheet models generally initialize at present day, or in the late 1990s.

**For future Copernicus manuscripts, consider putting the tables and figures inline rather than at the end. This makes the paper much easier to review and is allowed by Copernicus**

**as of January 2016.**
This is a great advice! We will do so in future Copernicus manuscripts.

**My recommendation is that the manuscript be published with minor corrections.**

**Specific Comments**
**p. 3 l. 11: You may wish to define SRES and RCP the first time you refer to them, though these acronyms will be familiar to most readers.**
We have now defined SRES and RCP the first time that we refer to them.

**p. 4 l. 26: Like Reviewer 1 (Christian Rodehacke), I felt that the XXX convention should be explicitly defined, even though it is likely obvious to the reader.**
We used Reviewer 1's suggestion to add the sub-clause "where XXX stands for different forcing scenarios as described later".

**p. 5 l. 2: It might be worth mentioning here that you will be discussing the method used to assess and evaluate the AGCM results in Sec. 4 (e.g. "...is to assess and evaluate (using metrics discussed in Sec. 4) CMIP atmosphere..."). During my first reading of the manuscript, I missed that the details of the analysis would come later (though you state it on p. 3 l. 19) and I was expecting at least some sense of what fields, metrics, etc. would be used in this analysis.**

Following a comment from V. Eyring, we have now moved up the description of the methods used to assess and evaluate AGCM and AOGCM results (initially in section 4) to this section.

**p. 8 l. 14: "The Tier 2 experiments..." You haven't yet introduced the tiers for the different experiments at this point in the text. If you can avoid referring to Tier 2 here by giving those experiments some other descriptor, that would save the awkwardness of needing to introduce the tiers here, rather than later where they seem to fit best.**

We have changed the sentence "The Tier 2 experiments" to "Another set of experiments". In addition, following a comment from the CMIP panel, we have now indicated the Tier for each experiment in the experiment tables.

**p. 8 l. 30-32: It is not entirely clear what "it" refers to in this sentence, presumably "accurate treatment of ice-ocean interactions"? More importantly, it seems to me that there is little doubt that accurate treatment of ice-ocean interactions requires moving boundaries in the ocean model. Just as parameterizing, rather than explicitly simulation, the circulation in ice-shelf cavities and resulting melt rates leads to inaccuracies, there can be little doubt that ignoring changes in cavity geometry (or parameterizing changes in melt rates) as the ice sheet evolves will lead to inaccuracies. All that is to say that "may" should be replaced with something stronger like "will likely".**

We have replaced "It may" by "Accurate treatment of ice-ocean interactions will likely".

**p. 9 l. 21-23: I would suggest moving "based on an initial analysis of AOGCM simulation[s] of ice sheet climate" to the beginning of the sentence for clarity. That way, it is hopefully clear that you are identifying the experiments based on the initial analysis, rather than that the ISMs are performing experiments based on the initial analysis. Also, maybe again here you could say that the criteria for determining which AOGCM results are "best" (i.e. chosen for the small subset of experiments) will be discussed in Sec. 4.**

We have modified the text as suggested.

**p. 10 l. 1-2: "...mismatch in spatial resolution over which SMB varies and that is used by AOGCMs". This phrase is confusing to me. Perhaps "..mismatch between the spatial resolution of AOGCMs and the characteristic length scale of variations in SMB"?**

We have modified the text as suggested.

**p. 10. l. 8: The use of RCMs as intermediaries between AOGCMs and ice-sheet models also adds ambiguity about which biases are introduced by the AOGCMs and which by the RCMs, does it not?**

Indeed, the use of RCMs introduces additional ambiguity about biases, and is a motivation for avoiding the use of RCMs. We have added a sentence to state this.

**Paragraph starting at p. 10 l. 26: Presumably, an effective melt parameterization would need to account for both the phenomena you outline in this paragraph (and probably more). It would need to make use of ocean temperature (and probably salinity) as a function of depth somewhere near the calving front each ice shelf and also the depth of the ice draft within the cavity. More sophistication would be nice (e.g. accounting for faster ocean flow with steeper ice-draft slope) but is still a topic of ongoing research.**

Indeed, we agree with the reviewer that an ideal effective melt parameterization would be more complex than the ones described in our manuscript. Given that it is still a topic of ongoing research, we are limited in our manuscript to propose solutions that have been used by the community. This said, ISMIP6 is engaging with the ice-ocean community (for example with the upcoming pre-AGU workshop) with the goal of identifying a better way to provide oceanic forcing.

The paragraph on the oceanic forcing has been expanded into four new paragraphs in light of the comments that you make below. The discussion now ends with "Ice-ocean interactions are an active area of research, and more complex parameterizations are being developed (e.g. Asay-Davis et al., 2016). ISMIP6 will organize workshops with the polar ocean community to investigate how to best derive oceanic forcing for ice sheet models, such that by the time the CMIP6 ocean models are evaluated, ISMIP6 may adopt a method that is distinct from those described above."

**p. 10 l. 27-28: I do not think that Rignot and Jacobs used surface temperature for their relationship, but rather ocean-bottom temperature close to the calving front. (However, they do not state their method of obtaining the temperature explicitly in their paper, at least as far as I could tell.) Also, this relation is only calibrated for melt rates at the GLs and likely is missing important nonlinearities (Holland et al. 2008).**

Thank you for pointing out the mistake of the use "surface" for the ocean temperature and the deficiency of the use of this simple parameterization. Indeed, Rignot and Jacob (2002) sentence "Delta $T$ is the difference between the nearest in situ ocean temperature measurement and the seawater freezing point (*43*) at a depth of 0.88 $H_p$ [Table 1, (*21*)]", which suggests an ocean bottom temperature. We have removed the word "surface" and added a sentence about the problem of using this relationship.

The new text now reads:

One possibility is to calculate melt rate anomalies from changes in the nearest ocean temperature using an observationally derived relation of 10 m yr$^{-1}$ °C$^{-1}$ (Rignot and Jacobs, 2002). However, this linear relation between ocean temperature and melt rates is calibrated for melt rates at the grounding line, and likely missing important non-linearities (Holland et al., 2008).

**p. 10 l. 29-30: "...that depends on the ocean temperature at the closest grid cell..." At what depth would the temperature be taken? Hopefully at or near the ocean bottom. Or better yet as a profile of depth.**

The parameterizations of Martin et al. (2011), Pollard and DeConto (2012), and DeConto and Pollard (2016) are all fairly similar and based on Beckman and Goose (2003). The oceanic melt rates are linked to ocean temperature using a relationship that takes the form of:
Melt = constant ( To – Tf) for Martin et al (2011)
Melt = constant ( To – Tf) | To – Tf | for Pollard and DeConto (2012), and DeConto and Pollard (2016)
Where To is a specified ocean temperature and Tf the ocean freezing point temperature at the ice shelf base, and the constant being a combination of density of the ocean water, density of the ice, specific heat capacity of ocean mixed layer, latent heat capacity of ice, thermal exchange velocity…

The specified ocean temperature, To, is different for each studies:
For Martin et al. it is set to -1.7C, a value that correspond to the Ross Ice Shelf from the work of Beckmann and Goose (2003), which according to Beckmann and Goose (2003) correspond to the "average temperature between 200 and 600m depth" for the Ross Ice Shelf.

For Pollard and Deconto (2012): "the ocean temperature To is specified differently for various Antarctic sectors, based on observations but mainly aiming to produce realistic ice shelf extents and grounding line position"

For DeConto and Pollard (2016), To is initially introduced in the method section as "ocean temperature interpolated from the nearest point in an observational (or ocean model) gridded dataset" and later defined as "the 1 degree resolution World Ocean Atlas temperatures at 400m depth"

We have turned the sentence in question into a paragraph to be more specific about the different approaches, and the various definition of ocean temperature. We also agree with the reviewer that ideally the ocean temperature would be a function of depth, and the revised manuscript reflects this.

The paragraph now reads:
"An alternative approach is to parameterize melt rates as proportional to the difference between ocean temperature at the shelf break and the freezing temperature at the ice shelf base. Beckman and Goosse (2003) developed such a scheme for ocean models, and similar schemes have been applied in offline ice sheet model simulations with idealized ocean forcing (e.g. Martin et al., 2011; Pollard and DeConto, 2012; DeConto and Pollard, 2016). In those studies, the ocean temperature is set to the average temperature between 200 and 600 m depth (Martin et al., 2011), or the temperature at 400 m depth (DeConto and Pollard, 2016), or specified differently for specific Antarctic sectors (Pollard and DeConto, 2012). Depending on the evaluation of the CMIP6 models, ISMIP6 may adapt one of these choices, or could prescribe depth-varying profiles of ocean temperature (and possibly salinity). The dependence of melt rates on thermal driving ranges from linear (Martin et al., 2011) to quadratic (Pollard and DeConto, 2012; DeConto and Pollard, 2016). Since the freezing temperature at the ice

base decreases with depth, the melt rates in all schemes tend to be higher near grounding lines, as found from observations."

**p. 10 l. 30-31: "If none of the CMIP6 ocean models are suitable" Can you be more specific about how "suitable" is defined (or refer to Sec. 4 and make sure you define there how you determine whether ocean results are suitable)?**

As mentioned in our earlier responses, the text that described the evaluation of CMIP6 models based on observations and regionally focused ocean models has now been moved ahead of this section. At a minimum, the CMIP6 ocean models will need to capture the broad scale polar ocean characteristics, and the ocean temperatures. However, at this time it is not possible to be more specific on a definition for "suitable", as the field is progressing rapidly, so new metrics may become available and the CMIP6 ocean models may have improved on the CMIP5 ocean models. We have replaced "suitable" by "can accurately capture the broad-scale polar ocean circulation or produce realistic near-shelf temperatures."

**p. 10 l. 31-32: "prescribe a melt parameterization that depends simply on the ice shelf draft". I (and other ocean modelers) feel that this is a poor choice (perhaps very much so) for a couple of reasons: 1) The thermal forcing (or thermal driving – the difference between the freezing point and the "ambient" ocean temperature, however "ambient" is defined) plays at least as important a role as the depth of the ice draft, so that differences between "warm" and "cold" ice shelves cannot be ignored. 2) Such parameterizations have only been used in small regions, where their coefficients have been calibrated to local thermal conditions, not over the whole of Antarctica.**

We agree that this is not the ideal choice, and that the difference between cold and warm shelves is important to capture. We have altered the manuscript to stress that if we have to use this method, the parameterization will not be uniform over the whole Antarctic, but will vary from one basin to the next, taking into account warm and cold shelves.

The manuscript now reads:

"If none of the CMIP6 ocean models can accurately capture the broad-scale polar ocean circulation or produce realistic near-shelf temperatures, an alternative is to prescribe a melt rate that simply depends on the ice shelf draft (e.g. Joughin et al., 2010a; Favier et al., 2014). This approach is less satisfactory, however, as it ignores temporal changes in ocean conditions, and typically uses coefficients calibrated to local thermal conditions. If ISMIP6 uses this approach, the provided coefficients would not be uniform, but would take into account that ocean waters reaching ice shelf cavities or fronts differ regionally. In Antarctica, for example, the ice shelves of Pine Island Glacier and Thwaites Glaciers lie in "warm" water, while the Filchner-Ronne or Ross ice shelves reside in "cold" water. Ocean temperatures reflect the dominant water sources, with warm waters dominated by circumpolar deep waters (Jacobs et al., 2011), while cold waters typically correspond to high salinity shelf water (Nichols et al., 2001)."

**p. 11 l. 1: "oceanic anomalies (basal mass balance and basal temperatures)" I do not see how these can be generated, independent of ice basal topography (ice draft) if a parameterization is being used. Instead, perhaps coefficients in the parameterization as functions of time could be provided, from which melt rates could be computed given an ice draft.**

This is a very good point, and parameterization that varies in time seems like a good idea, except that it might be difficult and time consuming for some groups to implement. At the same time, if our goal is to have all the models apply a similar anomaly (which will never be exactly the same as ice shelf areas vary from one model to the next), we will have to ignore the shelf draft and base it on something like the average depth between all the models, or the observed values. As stated above, the final decision on the oceanic forcing will be made after evaluation of the CMIP6

models, and after community workshops. The goal of the sentence was simply to state that we will distribute the forcing (or how to compute the forcing) via the ISMIP6 website. This sentence is not crucial to the text, so has been removed.

**Paragraph starting at p. 11 l. 20: I found this whole paragraph to be very confusing. Perhaps part of it is that initMIP has not yet been described. Maybe you could consider reordering the paper so initMIP has been described already at this point?**
We have reordered the manuscript so that initMIP is now already described, and slightly changed the wordings of the paragraph in response to your comments below and additional comments that we have received.

**p. 11 l. 21-22: It is not at all clear to me what these two sentences refer to. Is the 1990s to 2014 forcing repeated or is just 2014 repeated? Or something else?**
The forcing corresponds to the climate conditions at the end of the present-day initialization method. Present-day is defined differently for each model, and it also dependent on whether the initialization is an interglacial spin-up or whether it is mainly based on data assimilation (when data assimilation is used, present day also depend on what observations have been used). Therefore for one model, present day is year 1990, but for another model, present day will be year 2014. We have altered the manuscript to clarify these sentences.

**p. 11 l. 27: Please elaborate on the challenges of initializing ice sheet models to per-industrial conditions and how this presents challenges that do not allow for the typical historical run. This is likely not obvious to all of your readers.**
The quantity of accurate, high-resolution data available during the satellite era far exceeds that available for pre-industrial and historical periods. The majority of ice sheet models use these data in sophisticated initialization and assimilation procedures such that the present-day state of the ice sheets is simulated with a very high degree of fidelity. The lack of suitable data means that no such accuracy can be assumed for simulations of the historical periods. This becomes an issue because such inaccuracies are known to have a large effect on projections. For instance, discrepancies between projections can often be attributed to slight differences in the geometry of the ice-sheet margin assumed in a model (e.g., Shannon et al).

We have altered the manuscript to expand the discussion on the challenges of initializing ice sheet models to pre-industrial condition.

**p. 11 l 27-29: What does this mean? What period of time is covered? Please consider updating Table 1 so the range of times for the various ism simulations is given separately where they differ from the standard CMIP6 simulations. This would help to clarify the confusing differences in time ranges described in this paragraph compared with those of standard CMIP6.**
Because it is not possible for ice sheet models to initialize at a time that correspond to pre-industrial conditions (defined as year 1850 in the CMIP6 climate models) using data assimilation methods, the historical run for ice sheet models cannot start at year 1850. The historical run for ice sheet models can therefore only start from the "present-day" year that the ice sheet model was initialized at, which as clarified in an earlier response could be 1990, but also later. We hope that the rewritten version of this manuscript makes this clearer and we have updated the tables to show the distinctions between the start time of the CMIP6 and ISM runs.

**p. 13 l. 3-5: How will the abmb anomaly field be constructed? How will it be made to conform to differences in grounding lines and calving fronts between different models? Similarly, how will the SMB anomaly be made to conform to differences in ice sheet extent**

**between models in asmb?**

Because of the difference in ice shelf extent between the different models, the abmb anomaly is prescribed to be constant for each basin. This scalar value is different for each basin, and derived from the mean value of the ice shelf melt observed by Rignot et al. (2014) and Depoorter et al. (2014). The abmb anomaly is applied in the models everywhere where the ice is floating, so that the ice shelf area is the only parameter that impacts the amount of basal melt anomaly applied. For the SMB forcing, the procedure is the same for both Greenland and Antarctica. The schematic SMB anomalies are defined everywhere on the model grid and are therefore applicable for models with varying ice sheet extent.

We have modified the text to include this information.

**p. 13 l. 19-20: Why would it be ideal for the ism experiments to follow the AOGCM experiments with a six-month lag? Do you perhaps mean "no more than a six-month lag"? I would think ideally the ISM experiments would follow the AOGCM ones without any lag at all, but a realistic (or perhaps somewhat optimistic) time table would be for a six-month lag.** Indeed, ideally there will be no lag at all. However, we acknowledge that the climate modeling centers will be busy running simulations for other MIPs of CMIP6, hence we expect a time delay. We have however used your suggestion and rephrased so that the sentence now reads "Ideally, the XXX-withism and ism-XXX-self experiments would follow the corresponding AOGCM experiments with no more than a six-month lag."

**p. 15 l. 29-30: "regional ocean models (e.g. Timmermann et al. 2012)" FESOM, the model that was the primary focus of this Timmermann paper, is actually a global model with high resolution focused in Antarctica. Perhaps "regionally focused ocean models" would be more correct?**
We have modified the sentence as suggested.

**p. 33 Table 1: Please consider putting the actual start and end year for each ISM simulation that used a range different from the default CMIP6 (as requested previously)**
The start and end year for each ISM simulations have now been included in the table (now Table 2).

**p. 41 Figures A1: I feel this figure need some cleanup before they look professional enough for publication. The curves are not lined up very well (black is peaking out from under green). The blue arrows on the lighter blue surface and green base are not very visible. The giant gray error for freshwater flux should be given adequate room so it doesn't overlap the ice berg. Black lines should be anti-aliased and boundaries of the figure should not be jagged (slanted with respect to the figure caption).**
The figure has been cleaned up, and we are confident it is now ready for publication.

**p. 42 Figures A2: Blue text (both light and dark) is hard to read on blue background. The phrase "Liquid flux into the snowpack" should ideally either be entirely within or entirely outside the blue region.**
The figure has been cleaned up, and we are confident it is now ready for publication.

**Typographical Corrections**
**p. 5 l. 22: "amip" needs to be punctuated differently. Perhaps "The Atmospheric Model Intercomparison Project (amip; Gates et al. 1999) simulation allows..."**
We have done the suggested changes.

**p. 6. l. 18:** "four ISMs simulations" → "four ISM simulations"?
We have corrected the typographical mistake.

**p. 9 l. 22:** "AOGCM simulation" → "AOGCM simulations"
We have corrected the typographical mistake.

**p. 9 l. 28:** "...cannot be made however we list..." → "...cannot be made. However, we list..."
We have corrected the typographical mistake.

**p 10 l. 5:** RCMS → RCMs. Also, a verb is missing in "to SMB", perhaps, "to simulate SMB"?
We have corrected the typographical mistake, and added the missing verb, so it now reads "to simulate SMB".

**p. 10 l. 18:** "the SMB lapse rate obtained" I would remove the word "obtained". It is not needed.
We have removed the word "obtained" as suggested.

**p. 11 l. 8:** "the dynamic response output" I would remove the word "output".
We have removed the word "output" as suggested.

**p. 11 l. 11:** "Ice Sheet-Ocean-Model" → "Ice Sheet-Ocean Model"
We have corrected the typographical mistake.

**p. 11 l. 12:** "Sect. 3.3). )" there is some extra punctuation here. I think just the one end parenthesis is needed. Also, should this be Sect. 3.3.2?
We have corrected the typographical mistake (cleaned up extra punctuation and changed the mistake in the Sect reference, which should indeed have been 3.3.2).

**p. 11 l. 23-24:** I would change "our" to "the" at the beginning of both these sentences.
We have done the suggested changes.

**p. 11 l. 32:** "ice sheets evolution" → "ice sheet evolution"
We have corrected the typographical mistake.

**p. 12 l. 1:** "at least by 4 meters" → "by at least 4 meters"
We have corrected the typographical mistake.

**p. 12 l. 2:** "from the ism-lig-std" → "from ism-lig127k-std"
We have corrected the typographical mistake.

**p. 12 l. 30:** "Antarctic Ice Sheets" → "Antarctic Ice Sheet"
We have corrected the typographical mistake.

**p. 15 l. 28:** "As regional" → "Just as regional"
We have done the suggested changes.

**p. 19 l. 25-26:** "not explicitly asked to minimize the data request" → "not specifically requested as an output variable in order to reduce the size of the data files" (or something similar – the original phrasing is not very clear)
We have done the suggested rephrasing.

**p. 20 l. 26-39: Note that Asay-Davis et al. has been accepted in GMD so please don't forget to update the reference when the time comes.**
We have done the suggested update.

**p. 31 l. 5 and 18: In 2 references, Vizca.no is spelled without the accent mark while in three it is with the accent mark. This may be an issue with the respective journals but it looks strange when these articles are cited close together.**
We have done the suggested changes, namely to spell "Vizcaíno" consistently in both the manuscript and references.

**Reference**
**Holland, P. R., Jenkins, A., & Holland, D. M. (2008). The Response of Ice Shelf Basal Melting to Variations in Ocean Temperature. Journal of Climate, 21(11), 2558–2572. http://doi.org/10.1175/2007JCLI1909.1**

---

## Author Comment (AC5) · 2 Sep 2016

**Author's response to comments from C. Ritz (Comment 3, send by email via Copernicus)**

Thank you very much for having read the manuscript and for your suggestions. We have now revised our manuscript in light of these and other comments that we have received. A point-by-point reply is given below.

**The main points I have concern the initial state (spinup procedure)**
**1- As mentionned by reviewer 1 (p6L25) the initial state will be the same for AOGCM-ISM, XXX-withism and ism-XXX-self simulations and this initial state will come from the coupled AOGCM-ISM runs. I am afraid, this choice will prevent the groups that do not have the coupling working at the beginning of the intercomparison, to do XXX-withism and ism-XXX-self. Could you comment?**

We would first like to clarify that in our terminology "XXX-withism" stands for a specific AOGCM-ISM experiment. Therefore, we distinguish only two types of experiments (XXX-withism and ism-XXX-self ) for the point the reviewer is raising here.

XXX-withism requires the coupled spin-up to be done. If the spin-up and models were perfect, the initial states would be the same (see also response to reviewer 1, p6L25). The ism-XXX-self experiment is only meaningful in combination with a completed XXX-withism with the same combination of climate and ice sheet models, which implies the existence of the coupled initial state. Note also that XXX-withism is an ice sheet only experiment forced by AOGCM output and therefore likely easy to run, once the coupling technology has been developed for XXX-withism.

If a coupled spin-up is not available, we recommend doing the ism-XXX-self with what ever initial ice sheet state is available and repeat the experiment at a later date, starting from the coupled spin-up version. Doing this would in fact be useful to investigate the concern of reviewer 1 (how much difference does the initial state makes). The steering committee had discussed the possibility of including this has an additional experiment. In the end it was decided against, in order to minimize the number of simulations requested from the modeling centers.

**2- Do you plan to take advantage of the results of the initMIP project to refine the initial state procedure or simply use these result to quantify the trends related to the spinup procedure**

Results of the initMIP project are expected to point to specific aspects of ice sheet initialisation that have a crucial impact on sea-level projections and may be improved. While the initialisation procedures used by the different participating groups are not prescribed by ISMIP6, it is expected that individual groups will take advantage of the initMIP results to improve their initialisation procedures.

We have included a sentence in the manuscript to clarify the intention of initMIP further.

**3- I found difficult to follow which initial state will be used in the various simulations, would it be possible to add a column in table 1 to clarify this point?**
The manuscript has been rewritten to clarify this, as indirectly asked from reviewers 1 and 2. The experiment description table now includes a column to indicate the initial state for each simulation.

---

## Author Comment (AC1)

**Author's response to comments from G. Durand (Comment 1)**

**I enjoyed reading this manuscript clearly describing ISMIP6 initiative. I would however suggest to add a table similar to table 2. I think, it would be appropriate to list the ice sheet models with the corresponding research groups who expressed an interest in participating in this MIP**

Thank you very much for having read the manuscript and for your suggestion. We have now added a table similar to table 2, which lists the ice sheet models and the corresponding research groups.

---

## Author Comment (AC4)

**Author's response to comments from V. Eyring (Comment 2)**

**Comments from CMIP Panel**
**The CMIP Panel is undertaking a review of the CMIP6 GMD special issue papers to ensure a level of consistency among the invited contributions, also in answering the key questions that were outlined in our request to submit a paper to all co-chairs of CMIP6- Endorsed MIPs. We very much welcome the important contribution from ISMIP6 to the CMIP6 special issue, below are a few comments:**

We thanks the CMIP Panel for taking the time to review the ISMIP6 paper and for their leadership of CMIP6. We have revised our manuscript in light of these comments, and other comments that we have received.

**Please consistently use the term 'CMIP6-Endorsed MIPs' when you refer to other MIPs that are endorsed by CMIP6.**
We have checked the manuscript to ensure a consistent use of the term CMIP6-Endorsed MIPs.

**Please ensure consistency of the experiment names and abbreviations with the CMIP6 overview paper (Eyring et al., 2016).**
We have checked the manuscript to ensure a consistent use of the experiment names and abbreviations with the CMIP6 overview paper.

**Please ensure that all ISMIP6 'experiment_ids' and 'sub_experiment_ids' are consistent with those used in the CMIP6 data request and experiment table, and with the CMIP6 terminology (see email exchange with Karl Taylor).**
We have checked the manuscript to ensure that the use of the ISMIP6 experiment_ids and sub_experiment_ids are consistent with those used in the CMIP6 data request, experiment table and terminology, and use the ones from the email exchange with Karl Taylor.

**Please ensure that the 'source_id' for the offline models is compliant with CMIP6 terminology.**
We have emailed Karl Taylor for a clarification on what the source_id should be for the offline models. The response is that as we are not naming specific models, source_id is not relevant in our manuscript. We will however work with Karl Taylor to obtain source_id for the offline models that are compliant with CMIP6 terminology

**p.5,l1ff. Section 3.1**
**• Is there a more intuitive title that actually describes what is envisaged here? Use seems vague.**
We have changed the original title of section 3.1 from "Use of selected AGCM and AOGCM CMIP6 experiments" to "Analysis of experiments with climate models proposed elsewhere in CMIP6 (and not coupled to ISMs)". For consistency in the manuscript, we have also renamed section 3.2 from "Coupled AOGCM-ISM experiments" to "Experiments with climate models coupled to ISMs" and section 3.3 from "Standalone ice sheet model experiments" to "Experiments with ISMs not coupled to climate models".

**• The last two paragraphs (l21ff and p6,l1ff) on the definition and explanation of the DECK, CMIP6 historical simulation, ScenarioMIP, and PMIP experiments should be deleted since they are already defined elsewhere in this special issue. It seems sufficient to simply refer to the other papers.**
The last two paragraphs of this section are indeed intended to give a brief overview of CMIP,

CMIP6, ScenarioMIP and PMIP, for readers that are unfamiliar with these efforts. We do provide reference to these description papers, however, we do not want to delete these two short paragraphs, as they provide the context for the ISMIP6 experiments. In particular, the ice sheet modeling community is not always familiar with the CMIP effort, as it is the first time that there is an ice sheet component in the CMIP effort. Some of the ISMIP6 standalone experiments may appear strange for an ice sheet modeler, who does not know of the CMIP framework/history.

**• Would it be possible to list some observations that could be used to assess the models (l15ff)?**
The observations that could be used to assess the climate models were discussed in section 4.1. Following a comment of reviewer 2 that he expected to read something about the observations in this section, and your suggestion below that the first paragraph of section 3.1 could be merged with section 4.1, we have now moved the relevant paragraphs of section 4.1 to section 3.1. The moving of paragraphs required some tidying up of the text, which we have also done.

**• Possibly merge the first paragraph with Section 4.1 which is on the same subject and seems repetitive as it stands now with bits of information scattered across these two sections.**
Done, see comment above.

**p7,l19ff: please add to the text that both the downscaling method and the spin-up should be well documented. Also it might be good to expand a bit on the methods used for the spinup.**
We have added to the text that downscaling methods and spin-up should be well documented, as you suggested. We have also expanded a little on the methods and challenges for coupled AOGCM-ISM and ISM spin-up.

**p8,l8: 'then holds concentrations fixed for an additional two to four centuries.'. This is not how the 1ptCO2 experiment is defined. Note that in contrast to previous definitions, the experiment has been simplified so that the 1% CO2 increase per year is applied throughout the entire simulation rather than keeping it constant after 140 years as in CMIP5, see Section A1.4 in Eyring et al. (2016).**
When working with Karl Taylor on the experiment_id (after the paper was submitted) it became clear that the version of the 1pctCO2 experiment that ISMIP6 is using is in line with the CMIP5 version and thus slightly different from the CMIP6 version after 140yrs. We have modified the manuscript to state this.

**p8,l21: Here you encourage the extension of the projections until 2300 which is certainly a valid addition when it comes to the assessment of future sea level change. It is however only recommended, whereas in Table 1 this experiment is listed until 2300. Please could you make this consistent, e.g. either make it mandatory or - similar to ScenarioMIP (e.g. SSP5-8.5-Ext) - separate the two simulations in Table 1 into one that goes until 2100 and one that extends to 2300, and additionally list the Tier?**
Thank you for pointing this out. We will follow ScenarioMIP and have altered the manuscript to reflect this.

**p9,l5ff: Section 3.3: Could you confirm that all the output from the offline and standalone ISM experiments is conform to the output requirements of CMIP6? If ISMIP6 relaxes this requirement on output for some of its offline experiments, then those experiments should be considered not a part of CMIP6 (and therefore not listed in this paper). They could be described elsewhere. Our experience with past MIPs has been that initially the threshold effort required for standardizing data output (CMORization) is perceived as an obstacle by**

**many groups, but time and experience has shown that this effort is well worth it. We have found that only standardized data gets widely used by the community, and the analysis of that data, especially by researchers outside the major modeling centers, has been central to CMIP's success.**

We confirm that all the outputs from the offline and standalone ISM experiments will conform to the output requirements of CMIP6. We have worked with Martin Juckes, Denis Nadeau and Karl Taylor to test that CMOR works for polar stereographic grids. ISMIP6 will also continue to provide help to ice sheet modelers to make their files compliant, a process that started with the initMIP experiments**.**

**p9,l13ff: 'A key concern is that ISMIP6 assess uncertainty associated with both emission scenario and the AOGCMs' simulation of these scenarios. To this end, we anticipate identifying a subset of the CMIP6 AOGCM ensemble for use as ISM forcing which captures the full range of potential ice-sheet forcing.' This paragraph is too vague and the sentence seems contradicting - first a subset is selected and this should then be the full range? Please clarify. Please also add more explanation on how this selection process is done and why it is necessary.**

ISMIP6 seeks to assess the uncertainty in sea level projections arising from both the ice sheet models and the climate forcing. For a given emission scenario, the AOGCMs simulation of these scenarios will result in a range of atmospheric and oceanic forcings. It is not possible for the standalone ice sheet models to make simulations with the forcings from all AOGCMs. A subset will therefore be selected (rather than doing them all), and the subset will be chosen to represent the range, by using metrics of the SMB and ocean forcing to investigate that range. The manuscript has been modified to explain this.

**l18,p15: Data availability:**
**• Please delete 'the majority' in the first sentence. All CMIP6 simulations will be distributed through the ESGF and non-CMIP6 experiments shouldn't be described in this paper.**

We have deleted "the majority" as suggested, and checked that all experiments described in this paper are CMIP6 experiments.

**• Could you please add the following additional sentence after the first sentence? 'In order to document CMIP6's scientific impact and enable ongoing support of CMIP, users are obligated to acknowledge CMIP6, the participating modelling groups, and the ESGF centres (see details on the CMIP Panel website at http://www.wcrp-climate.org/index.php/wgcm-cmip/about-cmip).'**

We have added the suggested sentence.

**Table 1: The table lists experiments that are defined by ISMIP6 and experiments that are already defined elsewhere in CMIP6, this is confusing.**
**• Suggest to remove all experiments that are already defined elsewhere from this table (i.e., please remove the entire row for amip, abrupt-4xCO2, historical, ssp5-8.5, and lig127k).**
**• The titles of each category could be more specific by for example saying 'ISMIP6 DECK experiments' or something similar.**
**• Please could you also add a column that shows the Tier for each experiment?**
**• There could be another category that lists the ISMIP6 offline experiments from Section 3.3 if they are proposed to be part of CMIP6 (in which case the output has to be compliant with CMIP6 standards, see above).**

We have removed the rows corresponding to experiments that have already been defined in the CMIP6 description paper or MIP-Endorsed description papers of this special issue, and placed

them in a new Table (now Table 1). We feel that this information is important in our manuscript, as it allows readers that are not familiar with CMIP6 to avoid having to read many additional papers to find relevant information.

The new Table 2 now only contains experiments that are ISMIP6 experiments. This table includes your suggested changes: more specific titles, the Tier for each experiments, and the experiments that had been omitted from the original table (and described in the original section 3.3)

**Table 2:**
**• is it possible to add a column with some specifics on the ice sheet models used and a reference if available?**
**• GFDL and MPI-ESM were two more models that initially indicated interest in participating but are not listed?**
**• Except for CanESM that only participates in the diagnostic part of ISMIP6, are all other models listed using fully coupled ice sheet models, or are some of the models listed only contributing with standalone ice sheet models? Maybe this is not fully decided yet?**
**• Maybe it would be good to also list in a similar manner the standalone ice sheet models?**

The new Table 3 now includes the ice sheet models used. We have also added MPI-ESM (the omission had been a mistake on our part). However, GFDL informed us that they no longer wish to take part in ISMIP6 and have asked to be removed.

With the exception of CanESM, all the models listed in Table 3 are planning to participate using fully coupled ice sheet models.

The manuscript now includes a new Table (Table 4) that lists the standalone ice sheet models taking part in ISMIP6 (and their institution).

**Tables A1-A3: This is a very helpful overview of the variables requested by ISMIP6 but it would be good to clarify either in the caption or in a separate column from which CMIP6 experiments these variables are requested.**
The caption now indicates from which CMIP6 experiments these variables are requested.

**Table A2: Some additional information from the models is required to regrid the ocean data to standard grids. OMIP is proposing a weights file that model groups should provide to enable regridding from the native grid to one or two CMIP6 standard grids. Please refer to Griffies et al. (2016) and follow the same procedure for the ISMIP6 Omon requests if regridding is required.**
We have modified the manuscript as suggested.

**Reference:**
**Eyring, V., Bony, S., Meehl, G. A., Senior, C. A., Stevens, B., Stouffer, R. J., and Taylor, K. E.: Overview of the Coupled Model Intercomparison Project Phase 6 (CMIP6) experimental design and organization, Geosci. Model Dev., 9, 1937-1958, doi:10.5194/gmd-9-1937-2016, 2016.**

**Griffies, S. M., Danabasoglu, G., Durack, P. J., Adcroft, A. J., Balaji, V., Böning, C. W., Chassignet, E. P., Curchitser, E., Deshayes, J., Drange, H., Fox-Kemper, B., Gleckler, P. J., Gregory, J. M., Haak, H., Hallberg, R. W., Hewitt, H. T., Holland, D. M., Ilyina, T., Jungclaus, J. H., Komuro, Y., Krasting, J. P., Large, W. G., Marsland, S. J., Masina, S.,**

McDougall, T. J., Nurser, A. J. G., Orr, J. C., Pirani, A., Qiao, F., Stouffer, R. J., Taylor, K. E., Treguier, A. M., Tsujino, H., Uotila, P., Valdivieso, M., Winton, M., and Yeager, S. G.: Experimental and diagnostic protocol for the physical component of the  CMIP6 Ocean Model Intercomparison Project (OMIP), Geosci. Model Dev. Discuss.,doi:10.5194/gmd-2016-77, in review, 2016.

With many thanks for your ongoing efforts in the CMIP6 process.

The CMIP Panel